# Gas therapy potentiates aggregation-induced emission luminogen-based photo-immunotherapy of poorly immunogenic tumors through cGAS-STING pathway activation

Kaiyuan Wang [1,2,11], Yang Li [3,4,11], Xia Wang [5,11], Zhijun Zhang [6], Liping Cao [1], Xiaoyuan Fan [1], Bin Wan [1], Fengxiang Liu [1], Xuanbo Zhang [1,2], Zhonggui He [1], Yingtang Zhou [7] ✉, Dong Wang [6] ✉, Jin Sun [1] ✉ & Xiaoyuan Chen [2,8,9,10] ✉

The immunologically "cold" microenvironment of triple negative breast cancer results in resistance to current immunotherapy. Here, we reveal the immunoadjuvant property of gas therapy with cyclic GMP-AMP synthase-stimulator of interferon genes (cGAS-STING) pathway activation to augment aggregation-induced emission (AIE)-active luminogen (AIEgen)-based photo-immunotherapy. A virus-mimicking hollow mesoporous tetrasulfide-doped organosilica is developed for co-encapsulation of AIEgen and manganese carbonyl to fabricate gas nanoadjuvant. As tetra-sulfide bonds are responsive to intratumoral glutathione, the gas nanoadjuvant achieves tumor-specific drug release, promotes photodynamic therapy, and produces hydrogen sulfide ($H_2S$). Upon near-infrared laser irradiation, the AIEgen-mediated phototherapy triggers the burst of carbon monoxide (CO)/$Mn^{2+}$. Both $H_2S$ and CO can destroy mitochondrial integrity to induce leakage of mitochondrial DNA into the cytoplasm, serving as gas immunoadjuvants to activate cGAS-STING pathway. Meanwhile, $Mn^{2+}$ can sensitize cGAS to augment STING-mediated type I interferon production. Consequently, the gas nanoadjuvant potentiates photoimmunotherapy of poorly immunogenic breast tumors in female mice.

Cancer immunotherapy which drives patients' immune system to combat tumors has made great progress recently[1–3]. However, the anticancer efficacy of current immunotherapy is impeded by the immunosuppressive microenvironment in "cold" triple-negative breast cancer (TNBC), which typically characterizes deficient T cell infiltration and low immunogenicity[4]. Recently, more and more research works on phototherapy and nanotechnology have concentrated on affecting the immune system for cancer treatment[5]. Phototherapy, including photodynamic therapy (PDT) and photothermal therapy (PTT), has attracted increasing attention in cancer immunotherapy as it creates a tumor-associated antigen (TAA) pool and contributes to dendritic cell (DC) maturation[6,7]. Nonetheless, local phototherapy monotherapy is not enough to evoke strong and durable tumor immunogenicity for inhibiting tumor metastasis and rechallenge, it urgently needs to combine phototherapy and immunostimulation strategy to realize synergistic outcomes for photoimmunotherapy of poorly immunogenic TNBC[8].

Gas-based therapy, such as carbon monoxide (CO), nitric oxide (NO) and hydrogen sulfide (H₂S), is an emerging green therapeutic strategy for cancer treatment[9,10]. CO and H₂S gases are critical endogenous biosignaling molecules[11]. The delivery of exogenous CO/H₂S aims to induce mitochondrial membrane potential (MMP) depolarization, which would further damage mitochondria and lead to the release of mitochondrial DNA (mtDNA) into cytoplasm[12,13]. Here, we hypothesize that the existence of cytoplasmic mtDNA induced by gas therapy could activate cyclic GMP-AMP synthase-stimulator of interferon genes (cGAS-STING) pathway to exert immunoadjuvant effect[14]. The cGAS-STING pathway activation can trigger proinflammatory cytokine expression and type I interferon (IFN) response for immune stimulation and further augment the therapeutic effects[15]. Besides, manganese (Mn²⁺) has been reported to be responsible for the activation of the cGAS-STING pathway in various aspects, from promoting cyclic GMP-AMP (cGAMP) production to enhancing cGAMP-STING binding affinity[16,17]. In this work, we explore the synergy of immunity enhancement between gas therapy and Mn²⁺-based cGAS-STING promotion in both tumor cells and DCs.

In view of the above considerations, a versatile aggregation-induced emission (AIE)-active luminogen (AIEgen) is tactfully synthesized as phototheranostic agent. It shows robust reactive oxygen species (ROS) and heat generation after laser exposure, and near-infrared-II (NIR-II) fluorescence signal, which could realize PDT/PTT synergistic therapy and NIR-II fluorescence imaging (FLI)/photoacoustic imaging (PAI)/photothermal imaging (PTI) multimodal imaging. Nevertheless, the therapeutic outcome of PDT is usually restricted by intracellular antioxidant glutathione (GSH)[18,19]. Therefore, engineering a nanoassembly with the capability of ROS production and endogenous GSH depletion can simultaneously magnify oxidative stress for more effective tumor treatment[20]. Moreover, owing to the restricted active radius and short lifespan of ROS, it is important to promote the internalization efficiency of nanoassembly for more effective PDT[21]. The design of surface topography presents a perspective to increase the cell entry of nanoparticles. Recently, some researches have shown that the virus-like surface is beneficial in enhancing the nanoparticle-cell adhesive interaction and nanoparticle uptake efficiency compared to sphere-like surface[22,23].

In this work, the GSH/NIR sequentially initiated gas nanoadjuvant with AIEgen-mediated PDT/PTT synergistic therapy and tumor-specific amplified H₂S/CO/Mn²⁺ generation is constructed to effectively invade cancer cells and significantly activate cGAS-STING pathway for poorly immunogenic TNBC treatment. Specifically, the tetrasulfide-bridged virus-mimicking hollow mesoporous silica (tvHMS) is engineered to load AIEgen and manganese carbonyl (Mn₂(CO)₁₀, abbreviated as MnCO) (Fig. 1a). Following tumor accumulation, the virus-like surface helps gas nanoadjuvant invade cancer cells more effectively through spike surface-assisted adhesion. After entering the cancer cell, the overexpressed GSH could break the tetrasulfide bond to initiate the disintegration of nanoassembly, resulting in tumor-specific and controllable drug release, amplified PDT via GSH consumption, and H₂S production. Under NIR laser irradiation, the AIEgen-based PDT/PTT synergistic therapy could in situ activate the prodrug MnCO to generate Mn²⁺ and CO[24]. Both H₂S and CO stimulate mitochondrial dysfunction of tumor cells to induce the intracellular release of mtDNA, demonstrating the immunoadjuvant property of engaging cGAS-STING pathway. Moreover, Mn²⁺ is a powerful cGAS activator to enhance STING-mediated type I IFN response in both tumor cells and DCs, involving STING, tank-binding kinase 1 (TBK1), and interferon regulatory factor 3 (IRF-3) phosphorylation (Fig. 1b)[25,26]. Afterwards, the IFN-β expression level is improved, which contributes to DC maturation and cross-primes anticancer T cells for adaptive immune responses. As a result, TNBC suppression, potent distant tumor inhibition, and effective resistance to tumor metastasis and rechallenge are clearly manifested in various mouse tumor models. Integrating AIEgen-based phototherapy and gas-mediated immunostimulation strategy of cGAS-STING pathway activation into the virus-inspired gas nanoadjuvant illustrates a paradigm for photoimmunotherapy of poorly immunogenic tumor.

## Results

### Fabrication and characterization of gas nanoadjuvant

As a class of aggregation-induced emission (AIE) luminogens (AIEgens), 2, 2′-(2-((5′-(4-(diphenylamino) phenyl)-[2, 2-bithiophen]−5-yl) methylene)−1H-indene-1, 3(2H)-diylidene) dimalononitrile (TSSI) with exceptional ROS and heat generation capability has been prepared and thoroughly discussed in our previous work for photodynamic and photothermal therapy[27]. TSSI was purified with prominent yields of 80-85%, and the structure was verified using ¹H NMR[13],C NMR, and high-resolution mass spectra (HRMS) (Supplementary Fig. 1–3). Herein, we synthesized tetrasulfide-functionalized virus-like hollow mesoporous silica (tvHMS) nanoparticles for TSSI and manganese carbonyl (Mn₂(CO)₁₀, abbreviated as MnCO) encapsulation (MTHMS) (Fig. 2a, Supplementary Table 1). The tvHMS was fabricated via post-co-condensation of bis[3-(triethoxysilyl) propyl] tetrasulfide (TESPT, a tetrasulfide-functionalized silica precursor) with tetraethyl orthosilicate (TEOS) (v/v, 1:3) through the chemical homology principle on the solid silica surface. Firstly, the tetrasulfide-functionalized mesoporous silica-encapsulated solid silica (solid silica@tMS) nanoparticles were fabricated through post-condensation of TESPT and TEOS (v/v, 1:3) on solid silica surface, utilizing cetyltrimethylammonium bromide (CTAB) as the structure-directing agent[28]. Then, sodium hydroxide (NaOH) was used as the surface-morphological guide agent and etching agent for removing solid silica template and producing tvHMS. The pore-size of tvHMS was approximately 8.65 nm, determined by nitrogen adsorption-desorption isotherm, indicating the porous skeleton frameworks for cargo loading (Supplementary Fig. 4, 5)[29]. The underlying mechanism of the conversion of core-shell structure to the virus-mimicking hollow mesoporous structure by NaOH etching is as follows: (1) NaOH rapidly etches the solid silica to produce dissolved silicate species, the core is thus removed to yield the hollow structure; (2) NaOH slowly etches a part of mesoporous silica to produce dissolved silicate species, leaving behind the disordered, pot-holed surface layer while generating the spike structure. To further confirm the components of tvHMS, energy-dispersive X-ray spectroscopy (EDX) was used for elemental mapping study. As depicted in Fig. 2b, the Si, O, S, C, and N elements were present on the shell. The morphology of MTHMS was confirmed through the transmission electron microscope (TEM) (Fig. 2c), which showed a specific virus-like hollow core-porous shell structure with a spiky surface, with spike radial distance of ~20 nm. The dynamic light scattering (DLS) analysis demonstrated that mean hydrodynamic diameters of tvHMS and MTHMS were about 124.9 ± 5.2 nm and 128.1 ± 5.9 nm, respectively. The zeta potential of tvHMS and MTHMS were −17.8 ± 2.3 mV and −19.3 ± 3.2 mV, respectively (Fig. 2d). Moreover, MTHMS showed characteristic peaks of MnCO and TSSI in the UV-vis absorption spectrum (Fig. 2e), confirming the successful drug loading. The entrapment efficiency was calculated to 27.4 ± 2.8% and 32.6 ± 4.6%, respectively, according to the calibration curves of absorbance intensity from the UV-vis spectrum. Additionally, the photoluminescence (PL) spectra and photoacoustic signal further indicated the possibility of MTHMS for in vivo NIR-II FLI and PAI (Fig. 2f, g)[30,31]. The solution of MTHMS was then incubated in PBS or PBS + 10% FBS. At designated time point, aliquot of the solution was taken and monitored through DLS. As shown in Supplementary Fig. 6, the mean diameter of MTHMS in PBS or 10% FBS solution displayed negligible change after storage at ambient condition for 24 h, suggesting good in vitro stability of the MTHMS.

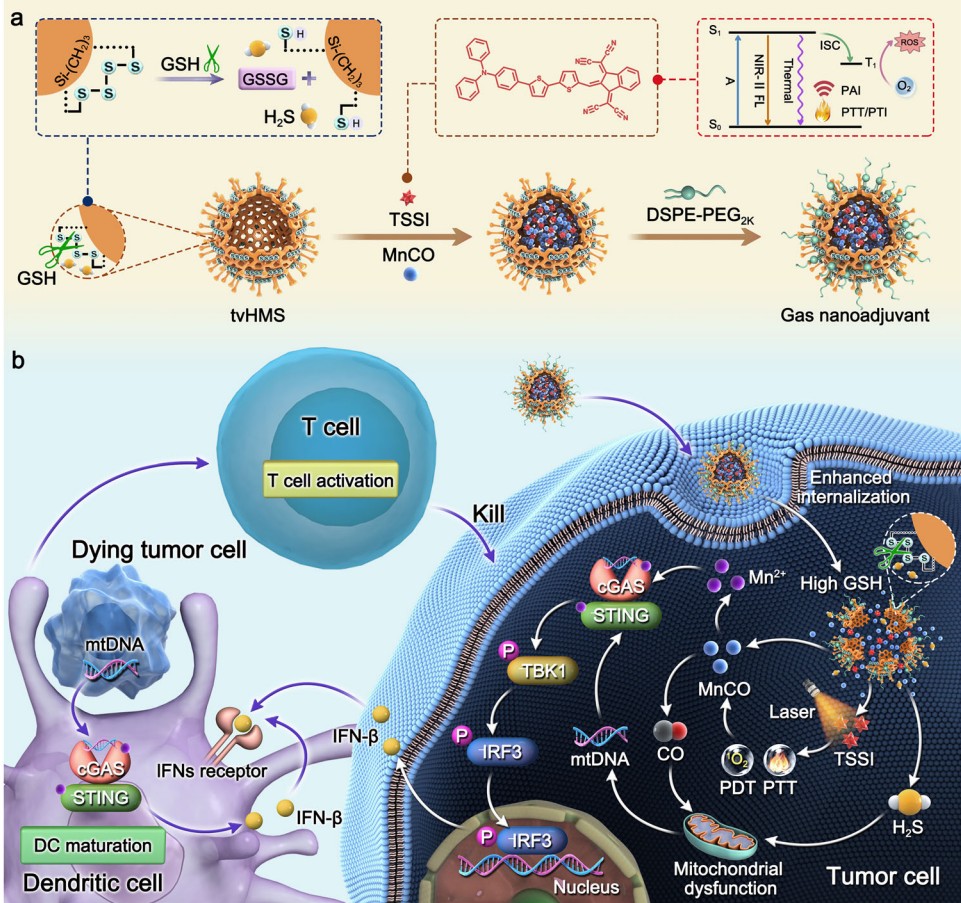

**Fig. 1 | The fabrication and biological functions of gas nanoadjuvant.**
**a** Schematic illustrating preparation routes for gas nanoadjuvant. **b** Schematic diagram of gas nanoadjuvant-based cGAS-STING pathway-dependent antitumor immune responses. Following tumor accumulation, the virus-like surface helps gas nanoadjuvant effectively invade cancer cell through spike surface-assisted adhesion. After entering cancer cell, the overexpressed GSH could break tetrasulfide bond that enables $H_2S$ generation and drug release. Upon NIR laser irradiation, the

AIEgen-based phototherapy could activate MnCO to produce CO and $Mn^{2+}$. Both $H_2S$ and CO stimulate the intracellular release of mtDNA to exert immunoadjuvant property by cGAS-STING pathway activation. Meanwhile, $Mn^{2+}$ is a powerful cGAS activator to enhance STING-mediated type I IFN response. The gas nanoadjuvant can engage cGAS-STING pathway in both tumor cells and DCs, leading to DC maturation and potent antitumor immune responses.

## Assessment of $H_2S$ generation and drug release

Because the tetrasulfide bond is sensitive to the reductive microenvironment, it makes a difference in enhancing PDT of TSSI by depleting GSH so as to induce more reactive oxygen species (ROS) generation and accumulation[32,33]. In order to ascertain the GSH consumption ability, the GSH level was investigated using 5,5′-dithiobis(2-nitrobenzoic acid) (DTNB), which can interact with free sulfhydryl[34]. As illustrated in Fig. 2h, UV-vis absorption spectrum exhibited a time-dependent loss of absorbance at 412 nm in tvHMS, indicating the clearance of GSH based on tetrasulfide cleavage. Given that the reaction between tetrasulfide bond and GSH could generate $H_2S$ for gas therapy, we evaluated the yield of $H_2S$ from tvHMS. As depicted in Fig. 2i, $144.5 \times 10^{-6}$ M of $H_2S$ could be detected from the mixture of 1 mg mL$^{-1}$ tvHMS and GSH at 48 h, and it increased to $199 \times 10^{-6}$ M when the tvHMS concentration reached 2 mg mL$^{-1}$. Moreover, Pb(NO$_3$)$_2$ soaked circular paper visualized the production of $H_2S$ from tvHMS. The specific reaction process was listed hereafter:

$$Si - (CH_2)_3 - S - S - S - S - (CH_2)_3 - Si + GSH \rightarrow GSSG + Si$$
$$- (CH_2)_3 - SH + H_2S \quad Pb^{2+} + H_2S \rightarrow PbS.$$

As shown in Fig. 2j, the color of test paper turned black and became deeper over time, suggesting sustained production of $H_2S$. In contrast,

the disulfide bond incorporated ones (dvHMS) did not change the color of Pb(NO$_3$)$_2$ soaked circular paper (Supplementary Fig. 7)[35].

Subsequently, the drug release profile of MTHMS was investigated. As shown in Fig. 2k, the cumulative release of TSSI was as low as $14.6 \pm 4.6\%$ after 48 h incubation without GSH, while the release was increased to $57.2 \pm 7.4\%$ and $88.6 \pm 7.6\%$ when GSH concentration was 5.0 mM and 10.0 mM, respectively. Moreover, TEM images revealed that an apparent change of MTHMS was found when incubated with pH 7.4 PBS containing 10 mM GSH (Fig. 2l). The particles appeared to swell and even collapse, manifesting MTHMS achieved rapid decomposition in tumor microenvironment (TME).

## Photothermal performance and production of ROS and CO

The ROS production capability of MTHMS was further assessed with 2,7-dichlorodihy-drofluorescein diacetate (DCFH-DA), which was converted to DCF to emit green fluorescence in response to ROS[36]. As shown in Fig. 2m, fluorescence signal at 525 nm of DCF increased rapidly with MTHMS under NIR irradiation, demonstrating the high ROS production efficiency. The heat production of MTHMS was subsequently evaluated under NIR irradiation (660 nm, 0.3 W cm$^{-2}$). As shown in Fig. 2n, o, a rapid rise of temperature was observed, which reached a plateau at 54 °C within 5 min, attributable to the active intramolecular motion of TSSI. The heat-producing ability makes MTHMS a desirable candidate for PTT. Meanwhile, the elevated

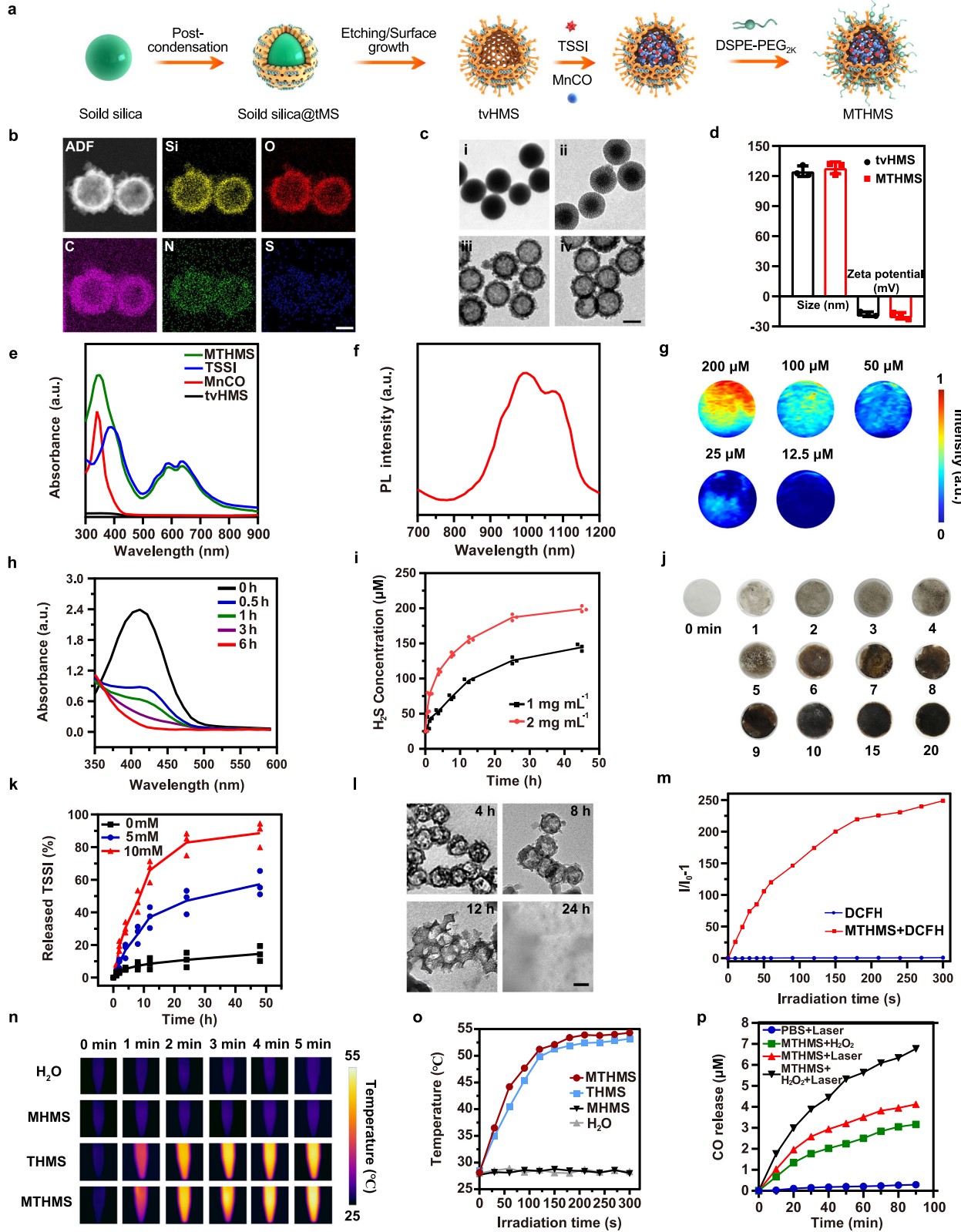

temperature is beneficial to the breaking of Mn-CO coordination bond to release extra CO.

As a CO releasing molecule (CORM), MnCO is widely explored[37]. To verify the phototherapy-triggered CO burst from MTHMS, the CO release was further quantified by hemoglobin (Hb) assay. It is worth noting that the combination of hydrogen peroxide ($H_2O_2$) and NIR irradiation (0.3 W cm$^{-2}$, 10 min) led to an obviously amplified

production of CO in MTHMS solution (Fig. 2p). This was ascribed to superior ROS and heat generation of MTHMS upon laser irradiation.

## Cellular uptake
The cellular uptake behavior of virus-like MTHMS was investigated in 4T1 cells, and the sphere-like MTHMS with similar surface charge and mean hydrodynamic diameter was chosen as a control (Fig. 3a, b,

**Fig. 2 | Preparation and characterization of MTHMS. a** Synthesis routes of MTHMS. **b** Representative EDS element mapping of tvHMS ($n = 3$ independent experiments). Scale bar, 100 nm. **c** Representative transmission electron microscope (TEM) images of (i) solid silica, (ii) solid silica@tMS, (iii) tvHMS, and (iv) MTHMS ($n = 3$ independent experiments). Scale bar, 100 nm. **d** Diameter distribution and zeta potential of tvHMS and MTHMS through DLS ($n = 3$ independent experiments). Data represent the mean ± SD. **e** Representative UV-vis spectra of tvHMS, MnCO, TSSI, and MTHMS ($n = 3$ independent experiments). **f** Representative PL spectra of MTHMS ($1 \times 10^{-5}$ M TSSI) ($n = 3$ independent experiments). **g** Representative photoacoustic images of MTHMS ($n = 3$ independent experiments). **h** Representative UV-vis spectra of DTNB for the detection of GSH depleting ability of tvHMS ($n = 3$ independent experiments). **i** $H_2S$ production from tvHMS dispersed in GSH solution ($n = 3$ independent experiments).

**j** Representative detection results of $H_2S$ production from tvHMS in GSH solution by $Pb(NO_3)_2$ soaked circular paper ($n = 3$ independent experiments). **k** Time-dependent TSSI release from MTHMS in GSH solution ($n = 3$ independent experiments). **l** Representative TEM images of MTHMS under GSH stimulus ($n = 3$ independent experiments). Scale bar, 100 nm. **m** Representative measurement of ROS production of MTHMS ($1 \times 10^{-6}$ M TSSI) under NIR irradiation ($n = 3$ independent experiments). **n** Representative thermographic images and (**o**) temperature change curves of $H_2O$, MHMS, THMS, and MTHMS ($1 \times 10^{-4}$ M TSSI) after NIR exposure (660 nm, 0.3 W cm$^{-2}$) ($n = 3$ independent experiments). **p** Representative measurement of CO generation at 37 °C after various treatments ($n = 3$ independent experiments). Source data underlying **d**–**f**, **h**, **i**, **k**, **m**, **o**, **p** are provided as a Source Data file.

Supplementary Fig. 8). From confocal laser scanning microscope (CLSM) observation, the uptake efficiency of MTHMS with virus-mimicking rough surfaces was much higher than that of sphere-like ones (Fig. 3c, d, Supplementary Fig. 9), confirming augmented entry of virus-mimicking MTHMS into 4T1 cells. The flow cytometry was employed for further quantitation analysis. As illustrated in Fig. 3d, Supplementary Fig. 9, virus-like MTHMS displayed 2.8 times higher uptake efficiency than spherical ones after 4 h incubation, suggesting that the virus-mimicking surface endowed MTHMS with superior cellular adhesion and internalization[38,39]. Compared to the smooth surface of sphere-like MTHMS, the rough surface endowed virus-like MTHMS with more contacting chances and elevated adhesion interaction with cytomembrane to improve the cellular internalization efficiency[39,40].

### Intracellular H2S/CO generation and mitochondrial dysfunction

The intracellular consumption of GSH and production of $H_2S$ after various treatments were monitored with ThiolTracker Violet and Washington State Probe-1 (WSP-1, $H_2S$ probe). As seen in ThiolTracker Violet fluorescence images (Supplementary Fig. 10), tvHMS-incubated 4T1 cells showed attenuated green fluorescence due to GSH depletion. As depicted in Fig. 3e, no apparent green fluorescence of WSP-1 was visible from cells incubated with PBS and disulfide bond incorporated virus-like hollow mesoporous silica (dvHMS). However, tvHMS treated group showed a notable enhancement of green fluorescence, indicating that the introduction of tetrasulfide bonds could initiate the GSH-induced intracellular generation of $H_2S$.

The intracellular generation of CO was further estimated through a CO detection system (FL-CO-1 + PdCl$_2$)[24,37,41,42]. As illustrated in Fig. 3f, MnCO encapsulated tvHMS (MHMS) treated cells showed higher fluorescence intensity compared with free MnCO treated cells, which may be attributed to the GSH depletion of tetrasulfide bond that facilitated $H_2O_2$-mediated MnCO activation. By contrast, more remarkable green fluorescence was emitted from MTHMS-treated cells after laser irradiation, confirming that the photodynamic and photothermal effect of TSSI could accelerate the breaking of coordination bonds in MnCO to release more CO.

$H_2S$/CO was reported to cause severe damage to cells with reduced mitochondrial membrane potential (MMP) concomitantly. Therefore, we monitored the mmP with JC-1 assay[43]. JC-1 polymerized to produce red fluorescence at high mmP but dissembled to a monomer yielding green fluorescence at low mmP. Compared to cells treated with TSSI + L ("+L" represents NIR laser irradiation), TSSI encapsulated tvHMS (THMS) + L treated cells displayed distinctly increased green fluorescence and clearly declined red fluorescence, suggesting significant mitochondrial dysfunction with the assistance of $H_2S$. Moreover, MTHMS + L treatment resulted in the highest green fluorescence intensity, which was ascribed to the synergetic effect of $H_2S$ and CO (Fig. 3g).

The DCFH-DA was utilized as an intracellular oxidant probe to measure ROS generation[44]. As shown in Fig. 3h, THMS + L group

presented significantly enhanced green fluorescence compared with TSSI + L, implying that the tetrasulfide bonds in tvHMS contributed to the depletion of GSH for improving the PDT efficiency under NIR exposure with enhanced ROS accumulation. Additionally, the fluorescence intensity in MTHMS + L group further elevated, which was attributed to the activation of mitochondrial ROS signaling pathway by CO-mediated gas therapy[45,46].

### mtDNA release and cGAS-STING pathway stimulation

To investigate whether $H_2S$/CO could induce mitochondrial DNA (mtDNA) release and further facilitate the cGAS-STING pathway activation in a synergistic manner, the cytosolic and extracellular mtDNA was investigated through quantitative reverse transcription polymerase chain reaction (RT-PCR)[47]. As seen in Fig. 4a, THMS + L group showed a higher level of cytosolic and extracellular mtDNA than TSSI + L group, confirming the $H_2S$-mediated mtDNA release. Notably, MTHMS + L group displayed much more mtDNA release due to the mitochondrial dysfunction synergistically induced by released CO. Furthermore, the cGAS-STING pathway existed in dendritic cells (DCs) and cancer cells. So, the levels of IFN-β, C-X-C motif chemokine 10 (CXCL10), and IL-6 in cancer cells after various treatments and bone marrow-derived dendritic cells (BMDCs) incubated with pretreated cancer cells were measured. As seen in Fig. 4b, c, the IFN-β, CXCL10, and IL-6 expression in MTHMS + L group elevated significantly both in cancer cells and BMDCs, evidencing the effective cGAS-STING pathway activation. The frequency of mature DCs induced by tumor cells with different treatments was evaluated because of cGAS-STING activation. 4T1 cells with different treatments were lysed and co-cultured with BMDCs. As expected, MTHMS + L treated cancer cells significantly boosted DC maturation (Fig. 4d, e, Supplementary Fig. 11), which was driven by the striking IFN-β release after cGAS-STING pathway activation. To further confirm the cGAS-STING activation-induced DC maturation, the highly potent and selective inhibitors (RU.521, C-178, or C-176) of the cGAS-STING pathway were separately added when incubating BMDCs with MTHMS + L treated cancer cells[48-50]. As shown in Supplementary Fig. 12, the cGAS-STING pathway inhibition significantly decreased the IFN-β, CXCL10, and IL-6 expression and reduced the frequency of mature DCs, evidencing the cGAS-STING pathway-dependent DC maturation.

### In vitro cytotoxicity and apoptosis assay

We first utilized the Chou-Talalay method to analyze the combination. The combination index (CI) was determined referring to the equation: $CI_n = (D)_a/(Dn)_a + (D)_b/(Dn)_b$, where $(Dn)_a$ and $(Dn)_b$ are the dosages of drugs a and b which suppress the cell growth by n%. And $(D)_a$ and $(D)_b$ express the respective drug concentrations of a and b in which the drug combo suppresses the growth of cells by n%. CI > 1, CI = 1, and CI < 1 represent antagonistic, additive, and synergistic effect, respectively. The growth inhibition effect (fraction affected, Fa) was determined referring to the equation: Fa = 1-(%growth/100). The plots of CI at various Fa levels were constructed. As depicted in Supplementary

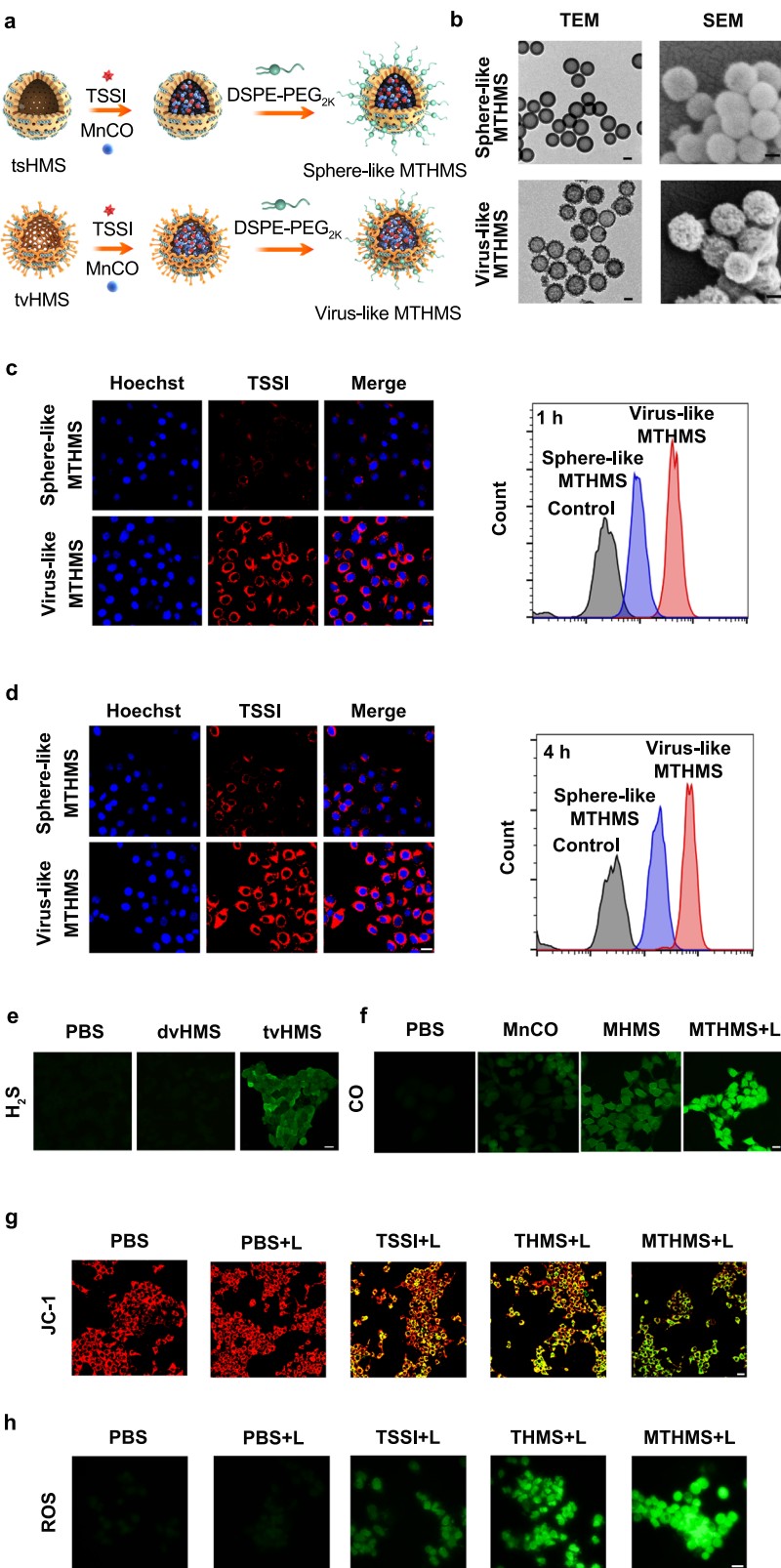

**Fig. 3 | Evaluation of MTHMS in vitro. a** Schematic diagram of fabrication and **b** TEM and scanning electron microscope (SEM) imaging of virus-like and sphere-like MTHMS (scale bar = 50 nm). CLSM observation and flow cytometric analysis of 4T1 cells incubated with virus-like and sphere-like MTHMS for 1 h **c** and 4 h **d** (scale bar = 15 μm). **e** WSP-1 ($H_2S$), **f** FL-CO-1 (CO), **g** JC-1 (red for aggregate and green for monomer), and **h** DCFH-DA (ROS) fluorescence imaging with various treatments (scale bar = 15 μm). For **b**–**h**, experiment was repeated three times independently with similar results.

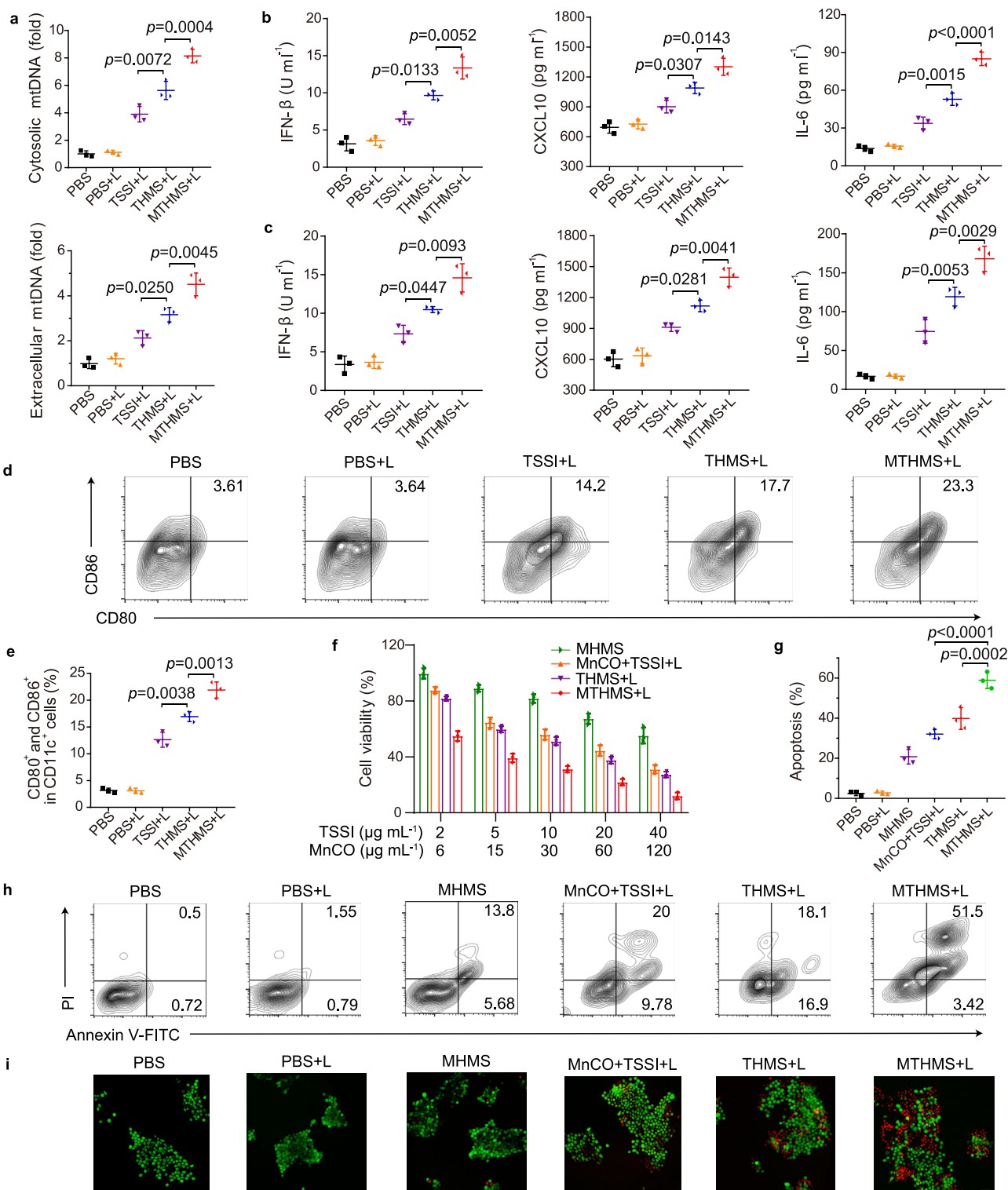

**Fig. 4 | Evaluation of gas nanoadjuvant-induced mtDNA release and antitumor immune response through cGAS-STING activation. a** RT-PCR detection of relative mtDNA copy number in cytosol and supernatant of 4T1 cancer cells after various treatments ($n$ = 3 independent experiments). The detection of cytokines (IFN-β, CXCL10, and IL-6) in culture supernatants of **b** 4T1 cells following the indicated treatments and **c** BMDCs incubated with pretreated cancer cells ($n$ = 3 independent experiments). **b** the $p$ values of MTHMS + L to THMS + L in the detection of IFN-β, CXCL10, IL-6 are 0.0052, 0.0143, and <0.0001, respectively. And the $p$ values of THMS + L to TSSI + L in the detection of IFN-β, CXCL10, IL-6 are 0.0133, 0.0307, and 0.0015, respectively. **d** Flow cytometric assessment images and **e** relative quantification of DC maturation (CD11c⁺CD80⁺CD86⁺) triggered by cancer

cells with different treatments ($n$ = 3 independent experiments). **f** Inhibitory effects of formulations on proliferation of 4T1 cancer cells investigated via CCK-8 assay ($n$ = 3 independent experiments). **h** Flow cytometric assessment images and **g** relative quantification of apoptosis of 4T1 cancer cells after receiving the indicated treatments, cells were stained with Annexin V-FITC/PI ($n$ = 3 independent experiments). **g** The $p$ values of MTHMS + L to THMS + L and MTHMS + L to MnCO +TSSI + L are 0.0002 and <0.0001, respectively. **i** Calcein-AM/PI staining of 4T1 cells with various treatments (scale bar = 40 μm). Experiment was repeated three times independently with similar results. Data represent the mean ± SD. Statistical significance was calculated through one-way ANOVA using a Tukey post hoc test. Source data underlying **a–c**, **e–g** are provided as a Source Data file.

Fig. 13, mass ratio 1:3 of TSSI to MnCO exhibits the best synergic effect under laser irradiation. The cytotoxicity of various treatments was then evaluated via CCK-8 assay. As shown in Fig. 4f and Supplementary Table 2, MHMS showed limited toxicity, the cell-killing rate was nearly $44.76 \pm 5.66\%$, even with a high concentration of MnCO at 120 μg mL$^{-1}$. While MnCO+TSSI + L and THMS + L exhibited apparently enhanced cytotoxicity against 4T1 cells with the survival rate of $31.17 \pm 3.02\%$ and $21.35 \pm 2.09\%$, respectively. Note that MTHMS + L treated cells displayed the lowest survival rate of $12.06 \pm 2.46\%$, which was owing to the synergistic antitumor efficacy of phototherapy and CO/H$_2$S gas therapy. The CI$_{50}$ value of MTHMS + L against 4T1 cancer cells was 0.32, demonstrating the strong synergism and good sequential MnCO activation. Moreover, the pretreated 4T1 cells were further incubated for 12 h, flow cytometry was adopted to test the cell apoptosis ratio after annexin V-FITC/PI staining (Supplementary Fig. 14). Up to $97.50 \pm 0.91\%$ of PBS treated cells were in the viable area (Fig. 4g, h). Nevertheless, the proportion of apoptotic and necrotic cells increased to $20.79 \pm 2.99\%$ and $39.83 \pm 4.43\%$ in MHMS and THMS + L treated groups. More evidently, extensive cell apoptosis and necrosis ratio ($58.91 \pm 3.36\%$) could be detected in MTHMS + L treated 4T1 cells, confirming the satisfactory therapeutic effect. In addition, the cell apoptosis and live/dead staining of various time points (0, 1, 3, 6, and 12 h) after laser irradiation were assessed to provide a time course for the cell death following MTHMS + L treatment. As shown in Supplementary Fig. 15a, b, the apoptosis ratio and cell death were gradually increased with prolonged incubation time after laser irradiation. Meanwhile, the live/dead staining further indicated the excellent antitumor potential of MTHMS + L (Fig. 4i).

## Multimodal imaging

TSSI possessed multimodal imaging capabilities including NIR-II FLI, PAI, and PTI, allowing to acquire rich and accurate tumor data[51]. As illustrated in Supplementary Fig. 16, a distinct NIR-II fluorescence signal was observed at the tumor site in MTHMS-treated mice compared with free TSSI-treated mice. The signal intensity gradually increased and peaked at 12 h post-injection. To further explore the biodistribution of MTHMS, the mice were euthanized to harvest tumors and main organs at 24 h post-injection for fluorescence imaging and quantitative analysis. As shown in Supplementary Fig. 17a, b, the tumor displayed higher fluorescence intensity than normal organs, probably driven by the enhanced permeability and retention (EPR) effect and favorable tumor adhesion as well as internalization based on the unique virus-like structure. Inspired by the exceptional efficiency of MTHMS to convert light to heat, the same tumor model was applied for PAI. As shown in Supplementary Fig. 18a, the photoacoustic signal in the tumor region significantly strengthened with time, and reached maximum at 12 h, which agreed well with the NIR-II FLI results. Afterwards, the PTI was assessed by monitoring the real-time temperature with an infrared thermal camera. As shown in Supplementary Fig. 18b, after being irradiated by 660 nm laser for 10 min, the temperature rised rapidly from 37.2 °C to 55.7 °C, reflecting the desirable potential of MTHMS for PTT application.

## In vivo anticancer study

We further evaluated the in vivo anticancer efficacy of tetrasulfide-doped virus-mimicking and GSH/NIR sequentially initiated gas nanoadjuvant. 4T1 tumor model was employed via inoculation of 4T1 cells into the mammary fat pad of BALB/c mice. On day 10, 4T1 cells were intravenously (i.v.) injected to simulate malignant tumor invasion and hematogenous metastasis (Fig. 5a)[52]. The additional i.v. injection of 4T1 tumor cells into tumor-bearing mice to simulate hematogenous metastasis has been widely applied as an artificial whole-body spreading tumor model[53–56]. Compared with spontaneous lung metastasis, the whole-body metastasis model was more aggressive and challenging, which was suitable for specialized anti-metastasis

evaluation. The circulating cancer cells can invade various organs, especially the lung. The treatment of the orthotopic tumor was done twice before the i.v. inoculation of the tumor cells. The tumor growth and body weight changes were recorded every two days. As seen in Fig. 5b–d, the tumor volume of saline group with/without laser irradiation presented a remarkable increase and reached ~1550 mm$^3$ on day 20. By contrast, MHMS exhibited a slight delay of tumor growth (~1160 mm$^3$). Moreover, THMS + L and MnCO+TSSI + L treatments showed distinct inhibition of tumor progression compared to the saline control, but still reached ~720 mm$^3$ and ~910 mm$^3$, respectively. Notably, the tumor growth was completely suppressed after MTHMS + L treatment (~260 mm$^3$). To evaluate the ROS level after MTHMS + L treatment, the DCFH-DA staining of tumor slides was performed. As seen in Supplementary Fig. 19, MTHMS + L treatment induced strong DCF fluorescence intensity in tumor tissue, implying intratumoral ROS burst. We then investigated the intratumoral CO level with FL-CO-1 fluorescence probe. As seen in Supplementary Fig. 20, MTHMS + L treatment provoked considerable CO generation in the tumor site. The survival curve was also recorded (Fig. 5e). As expected, mice in MTHMS + L group had a dramatically prolonged lifespan, displaying the highest survival rate (83%) within 40 days. Additionally, no pronounced reduction of body weight was detected in MTHMS + L group (Supplementary Fig. 21), implying no severe systemic toxicity. Meanwhile, the biosafety of MTHMS + L treatment was further validated by hematoxylin and eosin (H&E) staining of major organs and hepatorenal function measurement. As shown in Supplementary Fig. 22, 23, there was no visible organ damage in H&E staining and no noticeable abnormality in hepatorenal function. To confirm immunological memory initiated by MTHMS + L treatment, $2 \times 10^5$ 4T1 cells were injected into the left mammary fat pad of the survived mice in MTHMS + L group on day 40. Meanwhile, 4T1 cells were inoculated into naive mice as a control. As shown in Fig. 5f, g, the survived mice in MTHMS + L group showed effective resistance against cancer rechallenge, and the survival time was significantly prolonged, confirming that the gas nanoadjuvant could generate long-lasting antitumor immunity to suppress cancer relapse. Moreover, H&E-stained tumor slices of MTHMS + L treated mice showed serious injury. The Ki67 and terminal deoxynucleotidyl transferase-mediated deoxyuridine triphosphate nick end labeling (TUNEL) staining of tumor slices also confirmed the widespread apoptosis induced by MTHMS + L treatment (Fig. 5h). We next excised lungs for evaluating the degree of metastasis using Bouin's fluid staining. Compared with the saline group, no visible lung metastasis nodule could be observed in MTHMS + L group (Fig. 5i, Supplementary Fig. 24), demonstrating efficient suppression of lung metastasis. In addition, lung and liver metastases of tumor cells were assessed with H&E staining. As shown in Fig. 5i, the MTHMS + L treatment effectively inhibited the progression of tumor metastasis.

To investigate whether the observed antitumor effect is dependent on the immune system (on CD8 T cells), anti-CD8α antibody was intraperitoneally injected into the mice for in vivo lymphocyte depletion. The results showed that CD8$^+$ T cell depletion significantly impaired tumor suppression, led to severe lung metastasis, and shortened the survival time of mice treated with MTHMS + L (Supplementary Fig. 25a–e, 26, 27), confirming the central role of the immune system in gas nanoadjuvant-assisted photoimmunotherapy. Meanwhile, the survived mice in MTHMS + L group were also rechallenged with 4T1 cells with/without CD8$^+$ T cell depletion. As shown in Supplementary Fig. 25f, the tumor volume of CD8$^+$ T cell-depleted mice exhibited an evident increase and reached ~780 mm$^3$ on day 20, validating the effect of the immune system in resistance to tumor relapse.

We also evaluated the effect of MTHMS on tumor growth and lung metastasis without laser irradiation. The lack of laser irradiation resulted in the failure of PDT/PTT therapy and insufficient MnCO activation, which hindered the sequentially initiated stimulation of gas

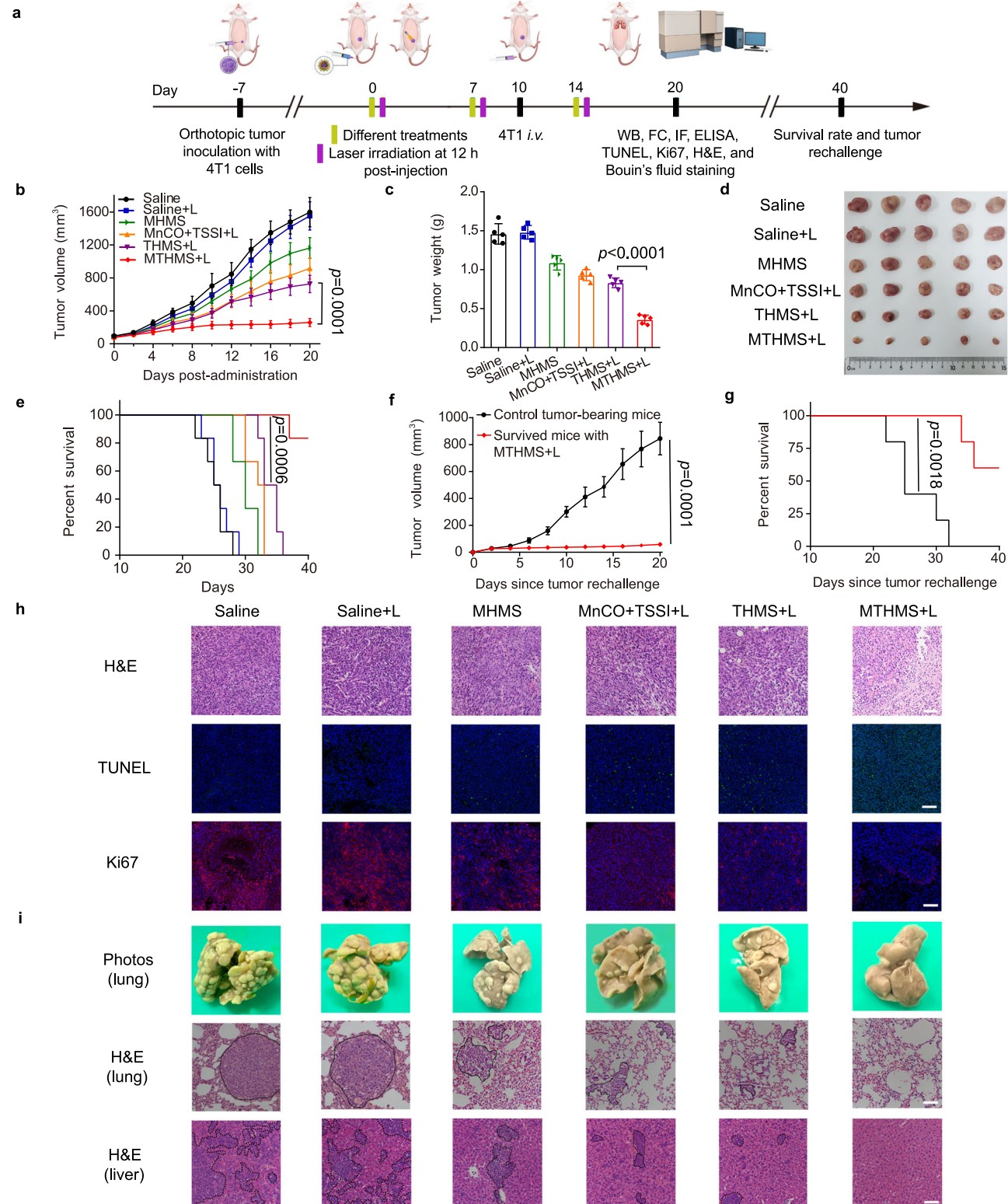

**Fig. 5 | Assessment of therapeutic efficiency of MTHMS. a** Schematic depicting 4T1 tumor model, 4T1 cells were intravenously (*i.v.*) administrated into the tumor-bearing mice on day 10 to simulate the hematogenous metastasis (WB: western blotting assay; FC: flow cytometry analysis; IF: immunofluorescence; ELISA: enzyme-linked immunosorbent assay; TUNEL: terminal deoxynucleotidyl transferase-mediated deoxyuridine triphosphate nick end labeling; H&E: hematoxylin and eosin). **b** Tumor growth curve (*n* = 5 mice), (**c**) tumor weight variations (*n* = 5 mice), (**d**) tumor photographs (*n* = 5 mice), and **e** survival curve (*n* = 6 mice) following different treatments. For (**c**), the *p* value of MTHMS + L to THMS + L is <0.0001. **f** Tumor growth curve and (**g**) survival curve of mice rechallenged with

4T1 cancer cells (*n* = 5 mice). **h** Representative images of H&E, TUNEL, and Ki67 staining of tumor slices collected from mice receiving various treatments (*n* = 3 mice). **i** Representative photos of lung stained with Bouin's fluid and representative images of H&E staining of lung and liver following different treatments (*n* = 3 mice). Dashed outlines indicate lung and liver metastases in H&E staining. Scale bar = 100 μm. Data represent the mean ± SD. Statistical significance was calculated through one-way ANOVA using a Tukey post-hoc test (**b**), log-rank (Mantel−Cox) test (**e**, **g**), or two-tailed student's *t* test (**f**). Source data underlying **b**, **c**, **e**–**g** are provided as a Source Data file.

nanoadjuvant. As shown in Supplementary Fig. 25a–e, 26, 27, MTHMS treatment displayed a slight delay of tumor progression (-1060 mm³) and a large number of lung metastasis nodules, indicating a significantly attenuated therapeutic effect.

### cGAS-STING pathway activation and antitumor immune responses

The released $Mn^{2+}$ in cytosol could stimulate cGAS and sensitize cGAS to leaked mtDNA induced by gas nanoadjuvant, promoting the activation of STING pathway and maturation of DCs (Fig. 6a)[57,58]. We further determined the cGAS-STING pathway-related protein expression and cytokine secretion in tumor tissue through western blotting assay and enzyme-linked immunosorbent assay (ELISA)[59,60]. As shown in Fig. 6b, in contrast to saline group with/without laser irradiation, the expression levels of phosphorylated STING (ᵖSTING), TBK1(ᵖTBK1), and IRF-3 (ᵖIRF-3) in MHMS group were much increased, implying that tumor microenvironment-driven $H_2S/CO/Mn^{2+}$ release could augment STING-mediated type I interferon production to some extent. Notably, MTHMS + L induced higher expression of STING pathway-related phosphorylated proteins in comparison with THMS + L, indicating that phototherapy-amplified MnCO activation provided the huge potential for cGAS-STING pathway activation. Meanwhile, the secretion of cGAS-STING-related cytokines such as IFN-β, CXCL10, and proinflammatory cytokines involving TNF-α, IFN-γ, IL-6, IL-12 were substantially increased in MTHMS + L group compared to the other groups (Supplementary Fig. 28), supporting the cGAS-STING pathway activation and the potential to switch the "cold" tumor to more sensitive "hot" tumor. We supposed that the MTHMS + L induced cGAS-STING pathway activation related to IFN-β secretion may contribute to DC maturation, which induced the subsequent T cell activation in tumor-draining lymph nodes (TDLNs). To test the above hypothesis, we explored the immune responses elicited by various treatments. As shown in Fig. 6c, Supplementary Fig. 29, the expression of mature DCs (CD11c⁺CD80⁺CD86⁺) in TDLNs significantly boosted after MTHMS + L treatment in contrast with the other groups. Mature DCs could present antigens to T lymphocytes to initiate tumor-specific adaptive immunity. Encouragingly, the frequency of tumor-infiltrating CD8⁺ T cells showed remarkable increase in MTHMS + L group (Fig. 6d, Supplementary Fig. 30). Consistently, the rate of tumor-infiltrating immunosuppressive regulatory T cells (Tregs, CD4⁺Foxp3⁺) in MTHMS + L group declined obviously in comparison with the other groups (Fig. 6e, Supplementary Fig. 30). Moreover, MTHMS + L treatment showed a marked limitation to anti-inflammatory M2-like tumor-associated macrophages (TAMs, CD206ʰⁱCD11b⁺F4/80⁺), while the population of proinflammatory M1-like TAMs (CD80ʰⁱCD11b⁺F4/80⁺) increased evidently (Fig. 6f, g, Supplementary Fig. 31, 32). Immunofluorescence staining visualized the noticeable increase of CD8⁺ T cell and macrophage infiltration in tumor tissue after MTHMS + L treatment (Supplementary Fig. 33), confirming the induction of robust immune responses. Furthermore, we monitored the level of CD8⁺ effector memory T cells (T_EM, CD3⁺CD8⁺CD62LˡᵒʷCD44ʰⁱ) of mice after different treatments. As seen in Fig. 6h, Supplementary Fig. 34, T_EM proportion in spleen was enhanced substantially after treatment with MTHMS + L, indicating a durable immunological memory built by the gas nanoadjuvant. To assess the tumor-specific T cell responses triggered by gas nanoadjuvant-assisted photoimmunotherapy, the percentage of gp70 tetramer-specific CD8⁺ T cells in the spleen was monitored by flow cytometry analysis. Following MTHMS + L treatment, the proportion of tumor-reactive gp70 tetramer-specific CD8⁺ T cells rose to a high level of ~4.5% (Supplementary Fig. 35, 36), indicating the induction of tumor-specific T-cell immunity.

### Abscopal effect in bilateral tumor model

We then constructed a bilateral tumor model and investigated whether gas nanoadjuvant could induce systemic immunity and suppress the

distant tumor[61,62]. The bilateral tumor model was established by injection of 4T1 cells into right mammary fat pad as the primary tumor on day 0. And on day 6, 4T1 cells were inoculated into left mammary fat pad of the same mice as the distant tumor. On day 7, when primary tumor reached ~80 mm³, mice were randomly assigned to two groups. Saline and MTHMS were intravenously injected on days 7 and 14. Only the primary tumor was irradiated directly at 12 h post-injection and the distant tumor was kept growing naturally (Fig. 7a). Encouragingly, MTHMS + L treatment displayed efficient suppression of both primary and unirradiated distant tumors (Fig. 7b–d), revealing an abscopal effect of gas nanoadjuvant-assisted photoimmunotherapy. To explore whether the tumor growth was affected by CD8⁺ T cells, anti-CD8a antibody was intraperitoneally injected into the mice for in vivo lymphocyte depletion. The results showed that CD8⁺ T cell depletion impaired the suppression of the secondary tumor after MTHMS + L treatment (Supplementary Fig. 37), confirming the critical role of CD8⁺ T cells in the reduced growth of the secondary tumor. To rule out the direct effects of the injected nanoparticles in the abscopal experiment, we evaluated the therapeutic outcome of MTHMS without laser irradiation. As shown in Supplementary Fig. 37, the volume of the secondary tumor increased significantly after MTHMS administration without laser irradiation, whereas MTHMS + L treatment effectively suppressed the secondary tumor, confirming the abscopal effect of gas nanoadjuvant-assisted photoimmunotherapy.

Immunosuppressive TME helps tumor progression and immune escape, which severely hinders antitumor activity. The inhibition of unirradiated distant tumor could demonstrate the activation of antitumor immune responses, which improves the recruitment of cytotoxic T lymphocytes and transforms immunologically "cold" TME to "hot" TME. The intratumoral infiltration of CD8⁺ T cells is a pivotal marker for defining "hot" and "cold" TME. As shown in Fig. 7e, MTHMS + L treatment caused much more CD8⁺ T cell infiltration in both primary and distant tumors, verifying the relief of immunosuppressive TME. Meanwhile, the remarkable decrease of Tregs in primary and distant tumors, as well as a significant boost of T_EM in spleen, was found in MTHMS + L group (Fig. 7f, g). We also evaluated the activation of tumor-specific T cell responses in the abscopal model. As shown in Supplementary Fig. 38, the MTHMS + L evoked the peak frequency of ~4.1% gp70 tetramer-specific CD8⁺ T cells, manifesting the elicitation of robust tumor-specific T cell immunity. Furthermore, MTHMS + L treatment led to high expression of proinflammatory cytokines in serum including TNF-α, IFN-γ, and IL-6 (Fig. 7h). In addition, no significant body weight change was monitored in MTHMS + L group (Supplementary Fig. 39).

## Discussion

TNBC characterizing significant morbidity and mortality is a common solid tumor malignancy. The immunosuppressive microenvironment in TNBC severely limits the efficacy of current immunotherapy. In the past few years, an increasing number of studies on phototherapy and nanotechnology have focused on affecting the immune system for cancer treatment. Phototherapy is a hopeful modality to eliminate cancer upon laser irradiation and meanwhile stimulates immune response for improved anticancer immunity. However, local phototherapy monotherapy is insufficient for the induction of durable and strong tumor immunogenicity to resist tumor metastasis and rechallenge. Therefore, it is of urgent need to develop a synergistic strategy to amplify the phototherapy-mediated immune activation for photoimmunotherapy of poorly immunogenic TNBC[63,64].

In recent years, diverse efforts have been devoted to eliminating tumors through the nanomaterial-based combination of phototherapy and gas therapy[65,66]. For example, Wan and coworkers loaded biocompatible L-arginine into PCN-224 as an NO donor to combine PDT and NO gas therapy[67]. Upon laser exposure, the donor L-arginine could react with ROS and $H_2O_2$ to produce NO with a wide diffusion range

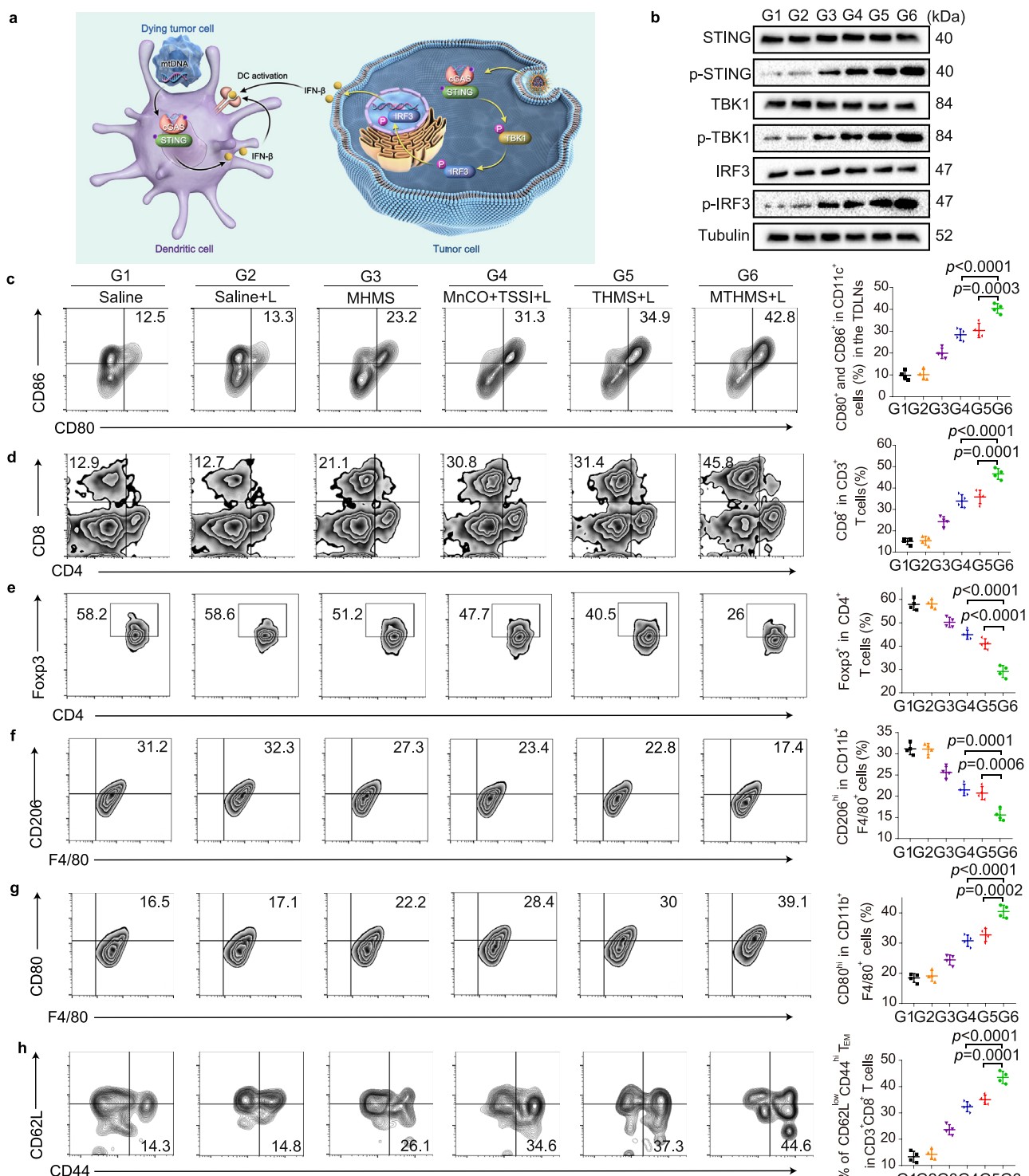

**Fig. 6 | The cGAS-STING pathway activation and antitumor immunity after treatments.** G1:Saline, G2:Saline+L, G3: MHMS, G4: MnCO+TSSI + L, G5: THMS + L, G6: MTHMS + L. **a** Schematic illustration of gas nanoadjuvant-mediated cGAS-STING promotion. **b** Western blotting assay of STING, TBK1, IRF-3, and phosphorylation of proteins in tumor. Samples derived from the same experiment and gels/blots were processed in parallel. Experiment was repeated twice independently with similar results. **c** Flow cytometric assay of mature DC (CD11c+CD80+CD86+) in TDLNs (*n* = 4 mice). The *p* values of G6 to G5 and G6 to G4 are 0.0003 and <0.0001. **d** Flow cytometric assay of tumor-infiltrating CD8+ in CD3+ T cells (*n* = 4 mice). The *p* values of G6 to G5 and G6 to G4 are 0.0001 and <0.0001. **e** Flow cytometric assay of tumor-infiltrating CD4+Foxp3+ Tregs (*n* = 4 mice). The *p* values of G6 to G5 and G6 to G4 are both <0.0001. Flow cytometric assay of tumor-infiltrating **f** M2-like macrophages (CD206hiCD11b+F4/80+) and **g** M1-like macrophages (CD80hiCD11b+F4/80+) (*n* = 4 mice). **f** *p* values of G6 to G5 and G6 to G4 are 0.0006 and 0.0001. **g** *p* values of G6 to G5 and G6 to G4 are 0.0002 and <0.0001. **h** Flow cytometric assay of CD3+CD8+CD62LlowCD44hi T_EM in spleen (*n* = 4 mice).The *p* values of G6 to G5 and G6 to G4 are 0.0001 and <0.0001. Data represent mean ± SD. Data were compared through one-way ANOVA using Tukey post-hoc test. Source data underlying **b**–**h** are provided as a Source Data file.

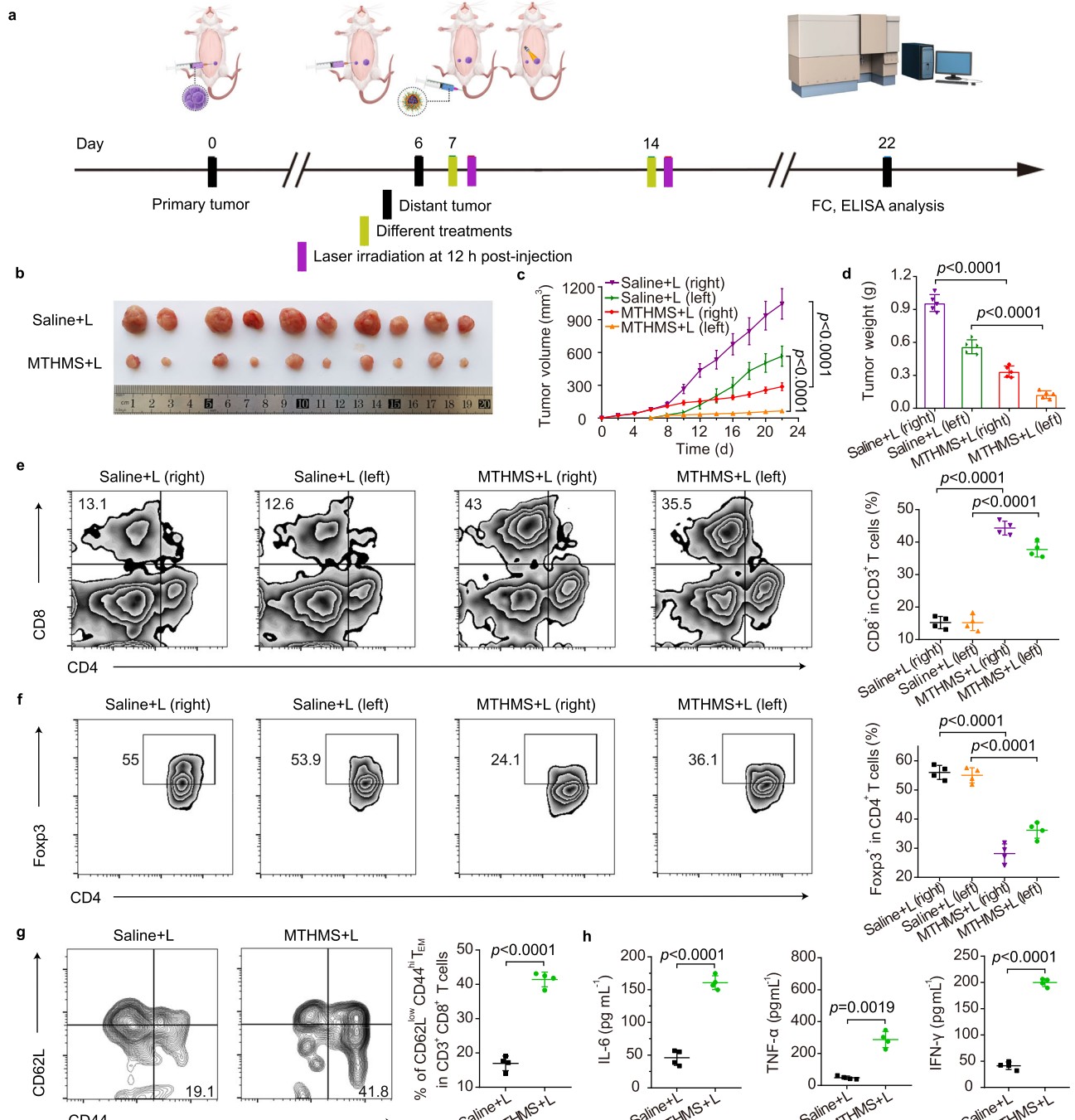

**Fig. 7 | Gas nanoadjuvant-induced systemic antitumor immunity. a** Schematic illustration of bilateral tumor model. Tumor on the right side represents "primary tumor" with laser irradiation, whereas tumor on the left side was defined as "distant tumor" without any treatment (FC: flow cytometry analysis; ELISA: enzyme-linked immunosorbent assay). **b** Tumor photographs, **c** tumor growth profiles, **d** tumor weight variations after the indicated treatments ($n = 5$ mice). **e** Flow cytometric assay and relative quantification of tumor-infiltrating CD8[+] in CD3[+] T cells ($n = 4$ mice). **f** Flow cytometric assay and relative quantification of tumor-infiltrating CD4[+]Foxp3[+] Tregs ($n = 4$ mice). **c–f** The $p$ values of MTHMS + L (right) to Saline+L (left) and MTHMS + L (left) to Saline+L (left) are all <0.0001. **g** Flow cytometric assay and relative quantification of CD3[+]CD8[+]CD62L[low]CD44[hi] $T_{EM}$ in spleen ($n = 4$ mice). **g** The $p$ value of MTHMS + L to Saline+L is <0.0001. **h** The secretion of cytokines (IL-6, TNF-α, and IFN-γ) in serum ($n = 4$ mice). **h** The $p$ values of MTHMS + L to Saline+L in IL-6, TNF-α, IFN-γ secretion are <0.0001, 0.0019, <0.0001, respectively. Data represent the mean ± SD. Statistical significance was calculated through one-way ANOVA using a Tukey post-hoc test (**c–f**) or two-tailed student's $t$ test (**g, h**). Source data underlying **c–h** are provided as a Source Data file.

and long half-life. In the hypoxic microenvironment, NO could sensitize cancer cells to PDT-generated ROS and almost completely eradicated cancer. However, the immunostimulating property of gas therapy was rarely explored to assist photoimmunotherapy.

The development of nanomedicine-based strategies for cGAS-STING pathway activation profoundly revolutionized cancer immunotherapy[68]. Recently, some DNA-damaging drugs, such as teniposide, cisplatin, and olaparib (PARP inhibitor), have manifested the capability to activate cGAS-STING signaling in cancer cells, inducing robust antitumor immune responses. For example, Hou and coworkers designed a nanoactivator that could lead to the presence of DNA in cytosol and improve $Mn^{2+}$ accumulation in cancer cells[59]. The

nanoactivators were stable in the systemic circulation and activated to release doxorubicin (Dox) and $Mn^{2+}$ to damage DNA and enhance cGAS-STING activity, which increased DC maturation and boosted intratumoral infiltration of cytotoxic T lymphocytes. Herein, we revealed the promising potential of tumor microenvironment-driven gas therapy in cGAS-STING pathway activation.

The prospective therapeutic implication of this work is the introduction of gas-mediated immunoadjuvant property into phototherapy to combat poorly immunogenic tumor. Gaseous molecule-based gas therapy shows great promise for antitumor treatment. Typically, both $H_2S$ and CO are critical physiological messenger molecules. The delivery of exogenous $H_2S$/CO could exert therapeutic effect by decreasing mmP, which would further damage mitochondria, and release mtDNA into cytoplasm. Thus, we explored whether the exposure of cytoplasmic mtDNA could stimulate immunity through cGAS-STING pathway activation, and revealed the immunoadjuvant effect of tumor microenvironment-driven gas therapy. Moreover, mesoporous silica has been widely applied in biomedicine and exhibited great biocompatibility. It was demonstrated that mesoporous silica is excreted through feces and urine. He and coworkers reported that urinary excretion could account for 15–45% of the injected mesoporous silica nanoparticles at 0.5 h post-administration[69]. The excellent biodegradability guarantees safety for clinical application.

In summary, the GSH/NIR sequentially initiated gas nanoadjuvant was developed via encapsulating AIEgen and MnCO into tetrasulfide-doped virus-mimicking hollow mesoporous silica. Under intratumoral GSH and in situ NIR sequential stimulation, the gas nanoadjuvant could achieve tumor-specific amplified $H_2S$/CO/$Mn^{2+}$ release to trigger immune responses by cGAS-STING pathway activation, which effectively assisted AIEgen-mediated PDT/PTT therapy in poorly immunogenic TNBC treatment. The synergistic therapeutic effect was verified through the enhanced tumor regression, effective distant tumor suppression, and significant elimination of tumor metastasis and rechallenge. This tetrasulfide-doped virus-mimicking and GSH/NIR sequentially initiated gas nanoadjuvant paves a strategy to boost photoimmunotherapy potency for poorly immunogenic tumor treatment.

## Methods

### Ethical statement
Our research complies with all relevant ethical regulations. All the animal protocols were performed in line with the Guidelines for the Care and Use of Laboratory Animals and approved by the Institutional Animal Ethical Care Committee (IAEC) of Shenyang Pharmaceutical University.

### Materials
Acetic anhydride and 1,3-bis(dicyanomethylidene)indane were bought from Meryer. Tetraethoxysilane (TEOS), bis[3-(triethoxysilyl) propyl] tetrasulfide (TESPT), cetyltrimethylammonium bromide (CTAB), and manganese carbonyl ($Mn_2(CO)_{10}$, abbreviated as MnCO) were bought from Sigma-Aldrich, USA. 1,2-distearoyl-sn-glycero-3-phosphoethanolamine-N-[methoxy (polyethylene glycol)−2000] ($DSPE-PEG_{2k}$) was supplied by Shanghai Advanced Vehicle Technology Pharmaceutical Ltd., China. Mitochondrial membrane potential assay kit (JC-1), 2,7-dichlorodihydrofluorescein diacetate (DCFH-DA), and cell counting kit-8 (CCK-8) were obtained from Dalian Meilun Biotechnology Co., Ltd., China. Lead nitrate ($Pb(NO_3)_2$), 5,5'-dithiobis (2-nitrobenzoic acid) (DTNB), ThiolTracker Violet, and Annexin V-FITC/PI kit were bought from Thermo Fisher Scientific. The calcein-AM/PI live/dead cell staining kit was acquired from Beijing Solarbio Science & Technology Co., Ltd, China. WSP-1 was purchased from Mkbio, China. RU.521, C-178, and C-176 were obtained from MedChemExpress, China. Mouse CXCL10, IFN-β, IL-12p70 and IL-6 enzyme-linked immunosorbent assay (ELISA) kits were supplied by NeoBioscience Technology Co, Ltd,

China. Mouse TNF-α and IFN-γ ELISA kits were acquired from Shanghai Jianglai Industrial Limited by Share Ltd, China. Scientz-IID Ultrasonic Homogenizer was obtained from Ningbo Scientz Biotechnology Co., Ltd, China. Round coverslips, cell culture dishes/plates, centrifuge tubes, and glass bottom culture dishes were purchased from Wuxi NEST Biotechnology Co., Ltd, China. All reagents and solvents utilized herein were of analytical standard grade unless otherwise stated.

### Cell lines and animals
4T1 cells (Serial: TCM32) were obtained from the Cell Bank of Type Culture Collection of Chinese Academy of Sciences (Shanghai, China). Cell line validation with short tandem repeat (STR) markers was conducted via Genetic Testing Biotechnology Corporation (Suzhou, China). In detail, eighteen STR loci were amplified using multiplex PCR. One additional marker (Human TH01) was used to screen for the presence of human species. The cell line sample was processed with ABI Prism 3130 XL Genetic Analyzer. Data were analyzed by Gene Mapper ID 3.2 software (Applied Biosystems). Appropriate positive and negative controls were run and confirmed for sample. The cell line tested negative for mycoplasma contamination. 4T1 cell line was maintained in RPMI-1640 medium supplemented with penicillin (100 U mL$^{-1}$), streptomycin (100 μg mL$^{-1}$), and 10% fetal bovine serum (FBS) in incubator with 5% $CO_2$ at 37 °C.

BALB/c mice (female, 6-8 weeks old) were supplied by the Animal Center of Shenyang Pharmaceutical University (Shenyang, Liaoning, China). Because the selected model was breast cancer, female animals were selected for experiments. The living environment of animals were maintained at a temperature of ~25 °C with a 12 h light/dark cycle, with free access to standard food and water. The humane endpoints included tumor burden exceeding 10% of normal body weight, animal weight loss exceeds 20% of normal animal weight, ulcer at tumor growth point, and sustained self-mutilation in animals. The humane end-point was approved by Certification and Accreditation Administration of the People's Republic of China (CNCA). Cervical dislocation under deep anesthesia was adopted for euthanasia. All animal procedures were carried out under the guidelines approved by the Institutional Animal Care and Use Committee (IACUC) of Shenyang Pharmaceutical University.

### Synthesis of TSSI
240 mg 1,3-bis(dicyanomethylidene)indane was added to the solution of 1 mmol TSSCHO in 10 mL acetic anhydride. The mixture was then stirred for 6.5 h under 60 °C. Following evaporating the solvent, washed the solid residue using diethyl ether (5 mL) and n-hexane (2 × 5 mL). The product was recrystallized from n-hexane/chloroform mixture to obtain TSSI. The structure was determined by $^1$H NMR[13],C NMR, and high-resolution mass spectra (HRMS)[1].H and $^{13}$C NMR spectra were recorded using Bruker ARX 400 NMR spectrometer. The NMR data were analyzed by MestReC 4.9.9.9 software. HRMS was obtained with Bruker ultrafleXtreme MALDI TOF/TOF mass spectrometer using FlexControl 3.4 software (Bruker Daltonics). Data analysis was conducted with FlexAnalysis 3.4 software (Bruker Daltonics).

### Fabrication of MTHMS
The tetrasulfide incorporated virus-mimicking hollow mesoporous silica (tvHMS) was synthesized as follows. The solid silica was fabricated through Stöber method. 0.4 mL $NH_3$·$H_2O$ and 1 mL TEOS was added to 22 mL water/ethanol mixture (2:20, v/v). Following stirring for 12 h, the product was gathered by centrifugation (7500 × $g$, 30 min), washed, and dried under vacuum. Next, 90 mg solid silica was added to 60 mL deionized water through ultrasonication. 25 mg NaOH and 700 mg CTAB were mixed in and stirred for 60 min under 60 °C. Afterwards, added the mixture of 16 mL cyclohexane and 4 mL TESPT/ TEOS (1:3, v/v) to the dispersion and kept stirring for 1 d. Followed by centrifugating (7500 × $g$, 30 min), washing and drying under vacuum

for 4.5 h, the tetrasulfide-functionalized mesoporous silica-encapsulated solid silica was fabricated. 180 mg tetrasulfide-functionalized mesoporous silica-encapsulated solid silica was then added to 18 mL 0.1 M NaOH and continued to stir for 2.5 h under 60 °C, the product was gathered through centrifugation (7500 × g, 30 min), washed using ethanol/water, and dried under vacuum for 4.5 h. Additionally, the TESPT was substituted with bis[3-(triethoxysilyl) propyl] disulfide (BTES) to fabricate disulfide bond incorporated ones.

For cargo encapsulation, 100 μL TSSI (10 mg mL$^{-1}$) and 350 μL MnCO (10 mg mL$^{-1}$) were added into 10 mL tvHMS (2 mg mL$^{-1}$) and kept stirring for 8.5 h. The products were acquired after centrifugation (7500 × g, 30 min) and washing by ethanol/water. To improve dispersibility of hollow mesoporous silica in the physiological solution, DSPE-PEG$_{2k}$ was anchored to their surface. Typically, nanoparticles and DSPE-PEG$_{2k}$ were dispersed in the ethanol solution under ultrasound assistance for 30 min and magnetic stirring overnight. The resulting MTHMS was gathered through centrifugation and was purified by washing with ethanol/water.

## Characterization of MTHMS

The entrapment efficiency (EE) of TSSI and MnCO was calculated in compliance with the formula below: EE (%) = (mass of loading content/mass of drug in-put) × 100%. The amount of loaded TSSI and MnCO was obtained by the established calibration absorption curve. The hydrodynamic size and zeta potential of MTHMS were measured via Zetasizer (Malvern, U.K.) using Zetasizer software 7.01. Scanning electron microscope (SEM) imaging was obtained through a SU-70 electron microscope. EDS element mapping and transmission electron microscope (TEM) imaging were performed by Talos F200 electron microscope. Pore-size distribution and nitrogen adsorption-desorption isotherm were recorded via the Micromeritics Tristar 3000 analyzer. UV-vis-NIR absorption spectrum was monitored using a UV-3600 spectrometer. Fluorescence spectra were determined utilizing Perkin-Elmer LS 55 spectrofluorometer. NIR photoacoustic imaging was recorded on VisualSonics Vevo-2100 system. The photothermal effect of MTHMS was detected through Fotric 226 thermal imaging system. The solution of MTHMS was incubated in PBS or PBS + 10% FBS. At designated time points (0, 2, 4, 6, 8, 12, and 24 h), an aliquot of the solution was taken and monitored through DLS to examine the in vitro stability.

## In vitro H$_2$S/CO release and ROS production

tvHMS was mixed with 10 mM glutathione (GSH) solution, and added 0.2 mM 5,5′-dithiobis(2-nitrobenzoic acid) (DTNB) for monitoring sulfhydryl at various timepoints (0, 0.5, 1, 3, and 6 h). The absorbance change at 412 nm was detected through UV-vis spectrometer. To quantitatively monitor H$_2$S production, 1 mg mL$^{-1}$ or 2 mg mL$^{-1}$ tvHMS was added to 10 mL GSH solution (10 mM) and stirred for various times (0, 0.5, 1.25, 3.75, 7.5, 12.5, 25, and 45 h). Next, the supernatant was gathered via centrifugation and mixed with 15 mL sodium acetate/zinc acetate mixture (mass ratio 1:4). Added 15 mg DMPD·2HCl and 25 mg FeCl$_3$ to form methylene blue. Following 15 min incubation, the absorbance at 660 nm was detected, and H$_2$S concentration was obtained via calibration curve.

To observe the production of H$_2$S, 15 mg tvHMS was added to 15 mL GSH solution (10 mM), a piece of 0.1 M Pb(NO$_3$)$_2$ soaked circular paper was then placed on the surface of the bottle. The color changes were observed at various times (0, 1, 2, 3, 4, 5, 6, 7, 8, 9, 10, 15, and 20 min). The disulfide bond incorporated virus-mimicking hollow mesoporous silica (dvHMS) was designed as a control.

The release of CO in PBS was determined via observing the transformation of UV-vis absorption spectra at 410 nm and 430 nm, which expressed the intensity of hemoglobin (Hb) and carboxyhemoglobin (HbCO), respectively. In brief, bovine Hb (4.2 μM) was added into pH 7.4 PBS (10 mM), then added excess sodium dithionite

(SDT) in a nitrogen atmosphere for Hb reduction. Subsequently, MTHMS was added into the above mixed solution and exposed to 660 nm NIR irradiation (0.3 W cm$^{-2}$) with or without hydrogen peroxide (H$_2$O$_2$). The UV-vis absorption spectra were recorded for the characterization of CO generation, the formula was shown below:

$$C_{CO} = \frac{C_{Hb} \times (528.6 \times I_{410\,nm} - 304 \times I_{430\,nm})}{216.5 \times I_{410\,nm} + 442.4 \times I_{430\,nm}} \tag{1}$$

The $C_{CO}$ and $C_{Hb}$ expressed concentrations of CO and Hb. $I_{410\,nm}$ and $I_{430}$ denoted absorbance at 410 nm and 430 nm.

The generation of ROS was assessed using DCFH-DA. Briefly, 0.5 mL ethanol containing DCFH-DA (1 × 10$^{-3}$ M) was added to 2 mL NaOH (1 × 10$^{-2}$ M) and stirred for 30 min. Then 10 mL PBS was applied to neutralize the hydrolysate. Afterwards, added the prepared DCFH-DA solution (4 × 10$^{-5}$ M) into MTHMS solution and illuminated the mixture with 660 nm irradiation (0.3 W cm$^{-2}$) for different lengths of time. The fluorescence at 525 nm was detected through PL instrument with excitation at 488 nm.

## In vitro drug release

The MTHMS with 0, 5, 10 mM GSH were wrapped into dialysis bags (MWCO = 14 000 Da) and soaked into PBS (60 mL pH 7.4) on a shaking bed under 37 °C (n = 3). Then withdraw 1 mL of solution outside the dialysis bags at selected time points for a quantitative analysis via UV-vis-NIR spectrophotometer. Meanwhile, 1 mL fresh medium was supplemented into the original medium. The release profile of TSSI was determined within 48 h.

## Cellular uptake

4T1 cells were inoculated into 12-well plates with 3 × 10$^4$ cells per well overnight. Then displaced the medium with fresh medium containing sphere-like MTHMS and virus-like MTHMS with the same concentration of TSSI at 10 μg mL$^{-1}$ for 1 h and 4 h. Thereafter, Hoechst 33342 was used to stain PBS washed cells for 10 min. After staining, washed the cells in PBS to elute the residual dye. Finally, the cells were collected for confocal laser scanning microscope (CLSM, C2SI, Nikon, Japan) observation with software NIS 4.13. For flow cytometric analysis with software BD CellQuest Pro, cells were processed following the same procedure without nuclear staining.

## Intracellular GSH consumption, H$_2$S/CO generation, and mitochondrial dysfunction

4T1 cells were inoculated into 12-well plates with 3 × 10$^4$ cells per well overnight. Then the medium was substituted with fresh RPMI-1640 containing different formulations for 6 h: (1) PBS; (2) tvHMS. After removing the residual nanomaterial, ThiolTracker Violet solution was added and incubated for 0.5 h. Subsequently, the cells were washed three times with PBS and an inverted microscope was used to obtain the fluorescence images.

4T1 cells were inoculated into 12-well plates with 3 × 10$^4$ cells per well overnight. Then substituted the medium with fresh RPMI-1640 containing different formulations for 6 h: (1) PBS; (2) MnCO; (3) MHMS; (4) MTHMS + L. The concentrations of MnCO and TSSI were 60 μg mL$^{-1}$ and 20 μg mL$^{-1}$, respectively. Next, added the mixture of PdCl$_2$ and FL-CO-1 (1:1) and incubated for 20 min, cells in group (4) were illuminated by 660 nm laser (0.3 W cm$^{-2}$) for 5 min. Subsequently, washed the cells three times with PBS and used an inverted microscope to obtain the fluorescence images.

To monitor the intracellular H$_2$S generation, 4T1 cells were incubated in fresh RPMI-1640 involving different formulations for 6 h: (1) PBS; (2) dvHMS; (3) tvHMS. After removing the medium, WSP-1 was added to co-culture for 0.5 h. Finally, cells were washed using PBS, and fluorescence imaging was performed with inverted microscope.

JC-1 probe was utilized to detect the depolarization of mitochondrial membrane. 4T1 cells were exposed to different treatments for 6 h: (1) PBS; (2) PBS + L; (3) TSSI + L; (4) THMS + L; (5) MTHMS + L. The concentrations of MnCO and TSSI were 60 μg mL$^{-1}$ and 20 μg mL$^{-1}$, respectively. Cells in groups (2), (3), (4), and (5) were irradiated with 660 nm laser (0.3 W cm$^{-2}$) for 5 min. Mitochondria were stained using JC-1 dye for 0.5 h, and potential change was measured based on the fluorescence images from CLSM (C2SI, Nikon, Japan).

As for the detection of ROS, 4T1 cells were incubated in fresh RPMI-1640 containing different formulations for 6 h: (1) PBS; (2) PBS + L; (3) TSSI + L; (4) THMS + L; (5) MTHMS + L. The concentrations of MnCO and TSSI were 60 μg mL$^{-1}$ and 20 μg mL$^{-1}$, respectively. Then replaced the culture medium with 1 mL serum-free medium involving 10 μM DCFH-DA and incubated for 0.5 h. After washing with PBS, cells in groups (2), (3), (4), and (5) were irradiated by 660 nm laser (0.3 W cm$^{-2}$) for 5 min and then incubated at 37 °C for another 30 min, followed by imaging with inverted microscope.

### Measurement of mtDNA release

Reverse transcription polymerase chain reaction (RT-PCR) was performed for quantitative detection of mtDNA release from mitochondria to cytosol and extracellular matrix. In detail, 4T1 cells were treated with: (1) PBS; (2) PBS + L; (3) TSSI + L; (4) THMS + L; (5) MTHMS + L. The supernatant was harvested to monitor the extracellular mtDNA. As for cytosolic mtDNA, the Mitochondrial DNA Extraction Kit (Phygene Scientific, PH1592) was applied for removing mitochondria from 4T1 cells. Next, Nucleic Acid Extraction Kit (Shanghai Biochip, SBC-025C) was used for extracting nucleic acid without mitochondria to quantitatively analyze cytosolic mtDNA. mtDNA ND1 (forward: 5'-CT CAACCCTAGCAGAAACAAACC-3', reverse: 5'-CGGAAGCGTGGATAAG ATGC-3') primer (Sangon Biotech) was utilized for RT-PCR. RT-PCR was conducted with Taqman Universal RT-PCR Master Mix (Applied Biosystems), primer (10 μM), and 0.4 μL probe (Sangon Biotech). The RT-PCR condition was set as below: activating Taq polymerase with 95 °C for 10 min, 40 cycles of 95 °C for 15 s and 60 °C for 60 s. Data were obtained at the end of cycle. Cytosolic mtDNA was normalized to GAPDH and analyzed with $2^{-\Delta\Delta Ct}$ method.

### cGAS-STING pathway stimulation in cancer cells and BMDCs

4T1 cells were inoculated into six-well plates and processed with: (1) PBS; (2) PBS + L; (3) TSSI + L; (4) THMS + L; (5) MTHMS + L. The cancer cell supernatant was collected for ELISA analysis to monitor CXCL10, IFN-β, and IL-6 concentrations. In addition, bone marrow-derived dendritic cells (BMDCs) were harvested from the tibias and femurs of BALB/c mice (female, 6–8 weeks old) and cultured in BMDCs medium: RPMI-1640 medium supplemented with 1% penicillin-streptomycin, 10% FBS, 10 ng mL$^{-1}$ IL-4, and 20 ng mL$^{-1}$ granulocyte-macrophage colony-stimulating factor (GM-CSF). Half replaced the culture medium every other day. On day 4, the gathered immature BMDCs were co-cultured with treated tumor cells for 24 h. For cGAS-STING pathway inhibition, BMDCs were separately pre-incubated with cGAS-STING inhibitors (RU.521, C-178, or C-176) for 1 h, and then co-cultured with MTHMS + L treated tumor cells without washing the inhibitors. The supernatants were harvested to measure CXCL10, IFN-β, and IL-6, and loosely adherent and non-adherent BMDCs were collected for flow cytometric analysis. Gating set up was based on fluorescence minus one (FMO) control.

### In vitro cytotoxicity

In vitro cytotoxicity was determined by CCK-8 assay. 4T1 cells were inoculated into 96-well plates ($5 \times 10^3$ cells per well) for 24 h. Subsequently, cells were processed with: (1) MHMS; (2) MnCO+TSSI + L; (3) THMS + L; (4) MTHMS + L. After 12 h incubation, cells in groups (2), (3), and (4) were illuminated with 660 nm irradiation (0.3 W cm$^{-2}$) for 5 min. Following further 12 h incubation, the medium was removed and

cells were washed three times using PBS. Then incubated the cells with serum-free medium involving 10% CCK-8 in the dark for 2 h. Eventually, the absorbance of product was detected at wavelength of 450 nm through the microplate reader (Thermo Scientific) with SkanIt software 2.4.3.37. The viability was presented as the percentage of absorbance value from experimental and control groups. The IC$_{50}$ value was calculated through GraphPad Prism 8.0. Furthermore, calcein-AM/PI Double Stain Kit was applied to stain dead and live cell at 0, 1, 3, 6, 12 h after laser irradiation, the fluorescence was observed via an inverted microscope.

### Apoptosis assay

Apoptosis assay was employed utilizing Annexin V-FITC/PI kit following the instruction. In brief, 4T1 cells inoculated into six-well plates were processed with: (1) PBS; (2) PBS + L; (3) MHMS; (4) MnCO+TSSI + L; (5) THMS + L; (6) MTHMS + L. After incubation for 12 h, cells in groups (2), (4), (5), and (6) were illuminated with 660 nm irradiation (0.3 W cm$^{-2}$) for 5 min. Following additional 0, 1, 3, 6, or 12 h incubation, 4T1 cells were digested using trypsin, resuspended into binding buffer, and added 5 μL Annexin V-FITC and 5 μL PI. Cell apoptosis was analyzed through flow cytometry. Gating set up was based on FMO control.

### Multimodal imaging

4T1 tumor-bearing mice were built based upon subcutaneous injection of $1 \times 10^6$ 4T1 cells in PBS buffer into the right flank of each mouse. Mice were randomly grouped. The mice were intravenously administrated with free TSSI solution and MTHMS at TSSI dosage of 2 mg kg$^{-1}$. NIR-II fluorescence imaging was performed on NIR-OPTICS Series III 900/1700 (China) with the long pass (LP) filter of 1000 nm, and PSViewer software 4.7.9 was employed to analyze the data. At 24 h after administration, tumor and main organs (heart, liver, spleen, lung, and kidney) were collected for fluorescence imaging. Photoacoustic imaging was recorded using the VisualSonics Vevo-2100 imaging system (Canada). Furthermore, in vivo photothermal imaging was carried out using Fotric 226 thermal imaging system upon 660 nm irradiation (0.3 W cm$^{-2}$) for 10 min at 12 h post-injection. Temperature variation was also recorded.

### In vivo anticancer study

Female BALB/c mice aged 6–8 weeks were applied and acclimatized to the animal center. For 4T1 tumor model, 100 μL cold PBS suspended with $1 \times 10^6$ 4T1 cells were injected into the right mammary fat pad of BALB/c mice. To simulate a more malignant invasion and hematogenous metastasis, we intravenously injected $5 \times 10^5$ 4T1 cells into the mice on day 10. All the tumor-bearing mice were randomly assigned into eight groups when the tumor volume calculated to be ~100 mm$^3$. On days 0, 7, 14, the mice were treated with saline, saline+L, MHMS, MnCO+TSSI + L, THMS + L, MTHMS + L + aCD8, MTHMS, MTHMS + L (2 mg kg$^{-1}$ TSSI, 6 mg kg$^{-1}$ MnCO), respectively. As for phototherapy, the tumor area was irradiated at 12 h post-injection using 660 nm laser (0.3 W cm$^{-2}$) for 10 min. For CD8$^+$ T-cell depletion, 100 μg anti-CD8α antibody (BP0061, BioXCell, clone 2.43) was intraperitoneally injected into the mice every four days since day 0. Body weights and tumor sizes were monitored every other day, and the survival rate was observed daily. The tumor volume was calculated referring to the equation: Volume = (Length × Width × Width)/2. Mice were euthanized and main organs (heart, liver, spleen, lung, and kidney) as well as tumors were harvested on day 20. The lung was stained with Bouin's fluid. Tissue slices were fixed for hematoxylin and eosin (H&E) staining, and tumors were stained by Ki67 and TUNEL. Blood samples were collected for assessment of hepatorenal function. The cytokine level of tumor tissue homogenate was monitored by ELISA referring to manufacturer's instruction. DCFH-DA and FL-CO-1 were used to detect the in vivo ROS generation and CO release after MTHMS + L treatment, respectively.

For tumor rechallenge study, $2 \times 10^5$ 4T1 cells were injected into the left mammary fat pad of survived mice pretreated with MTHMS + L on day 40. At the same time, equivalent 4T1 cells were injected into the untreated naive mice as a control. For the CD8[+] T cell depletion, 100 µg anti-CD8α antibody (BP0061, BioXCell, clone 2.43) was intraperitoneally injected into the survived mice in MTHMS + L group every four days since tumor rechallenge. Tumor volume and survival were recorded.

The bilateral tumor model was developed via injection of $3 \times 10^5$ 4T1 cells into right mammary fat pad as the primary tumor on day 0. On day 6, $2 \times 10^5$ 4T1 cells were inoculated into left mammary fat pad of the same mice as the distant tumor. On day 7, while primary tumor grew to ~80 mm³, the mice were divided into four groups randomly. On days 7, 14, mice were treated with saline+L, MTHMS + L + aCD8, MTHMS, and MTHMS + L ($2 \, mg \, kg^{-1}$ TSSI, $6 \, mg \, kg^{-1}$ MnCO), respectively. As for phototherapy, only the primary tumor was irradiated directly at 12 h post-injection using 660 nm irradiation ($0.3 \, W \, cm^{-2}$) for 10 min and kept the distant tumor growing naturally. For the CD8[+] T cell depletion, 100 µg anti-CD8α antibody (BP0061, BioXCell, clone 2.43) was intraperitoneally injected into the mice every four days since day 7. The tumor sizes as well as body weight variations were monitored every two days. Mice were sacrificed and tumors were collected on day 22. The cytokine level of serum was detected via ELISA following manufacturer's protocol.

## ELISA analysis

The standard or sample was added to each well and incubated for 90 min at 37 °C, then the wells were washed five times. Next, biotin-labeled detection antibody was added and incubated for 60 min at 37 °C, then the wells were washed five times. Subsequently, streptavidin conjugated HRP was added and incubated for 30 min at 37 °C, then the wells were washed five times. Afterwards, 3,3′,5,5′-tetramethylbenzidine (TMB) was added and incubated for 15 min at 37 °C. Finally, stop solution was added and monitored at 450 nm immediately.

## Western blotting assay

The proteins derived from 4T1 tumor tissues were extracted and further quantified by the BCA kit. Subsequently, the samples with equal amounts of proteins (20 µg) were separated through SDS-PAGE gel. Then the PVDF membranes with transferred proteins were blocked by 5% skim milk. After incubation with primary antibodies, including phospho-IRF-3 (Ser396) (4D4G) (4947 S, Cell Signaling Technology, 1:1000 dilution), IRF-3 (D83B9) (4302 S, Cell Signaling Technology, 1:1000 dilution), phospho-TBK1/NAK (Ser172) (D52C2) (5483 S, Cell Signaling Technology, 1:1000 dilution), TBK1/NAK (D1B4) (3504 T, Cell Signaling Technology, 1:1000 dilution), phospho-STING (Ser365) (D8F4W) (72971 S, Cell Signaling Technology, 1:1000 dilution), STING (A21051, ABclonal, 1:10000 dilution), and Tubulin β (bs-20694R, Bioss, 1:1000 dilution) overnight at 4 °C, the PVDF films were incubated with anti-rabbit IgG, HRP-linked antibody (7074 S, Cell Signaling Technology, 1:2000 dilution) for another 1 h. Chemiluminescence detection was carried out for protein band visualization with ECL Substrate. Uncropped and unprocessed full scan images of all western blots can be found in the Source Data file.

## Flow cytometry

Tumors isolated from mice were separated into small sections. The tumor sections were homogenized in staining buffer with digestive enzymes (collagenase IV, hyaluronidase, and deoxyribonuclease I) to acquire single cell suspension. Next, cells were stained with fluorescence-labeled antibodies, F4/80 (Biolegend, cat. no. 123116, Clone: BM8, 1:200 dilution), CD11b (Biolegend, cat. no. 101208, Clone: M1/70, 1:200 dilution), CD206 (Biolegend, cat. no. 141716, Clone: C068C2, 1:200 dilution), CD80 (Biolegend, cat. no. 104722, Clone: 16-

10A1, 1:200 dilution), Foxp3 (Biolegend, cat. no. 126404, Clone: MF-14, 1:200 dilution), CD4 (Biolegend, cat. no. 100432, Clone: GK1.5, 1:200 dilution), CD3 (Biolegend, cat. no. 100204, Clone: 17A2, 1:200 dilution), CD8 (Biolegend, cat. no. 100712, Clone: 53-6.7, 1:200 dilution) following instructions provided by the manufacturer. Eventually, flow cytometer (BD FACSCalibur) was applied to monitor the stained cells and FlowJo software was employed to analyze the data. Gating set up was based on FMO control.

The extraction of lymph nodes was carried out with surgical equipment, followed by crushing between the surface of frosted microscope slides into well containing PBS. Next, the cell mixture was filtered using 70 µm cell strainer into the conical tube. Afterwards, stained the cells with fluorescence-labeled antibodies, CD80 (Biolegend, cat. no. 104707, Clone: 16-10A1, 1:200 dilution), CD86 (Biolegend, cat. no. 105011, Clone: GL-1, 1:200 dilution), CD11c (Biolegend, cat. no. 117306, Clone: N418, 1:200 dilution) per manufacturer's procedure. Stained cells were analyzed using flow cytometer (BD FACS-Calibur) and evaluated using FlowJo software. Gating set up was based on FMO control.

Spleen was surgically harvested with sterilized surgical equipment. Spleen mixture was filtered into the 50 mL conical tube by a filter, and centrifuged at $500 \times g$ for 5 min. Following washing the mixture, cell pellet was resuspended using red blood cell lysis solution for 5 min. Cells were then stained by fluorescence-labeled antibodies, CD3 (Biolegend, cat. no. 100218, Clone: 17A2, 1:200 dilution), CD8 (Biolegend, cat. no. 100706, Clone: 53-6.7, 1:200 dilution), CD62L (Biolegend, cat. no. 104428, Clone: MEL-14, 1:200 dilution), and CD44 (Biolegend, cat. no. 103008, Clone: IM7, 1:200 dilution) referring to manufacturer's protocol. Stained cells were monitored using flow cytometer (BD FACSCalibur) and evaluated using FlowJo software. Gating set up was based on FMO control.

The tetramer staining analysis through peptide-MHC tetramer tagged using PE was performed to investigate the percentage of antigen-specific CD8[+] T cells. In brief, splenocytes were resuspended with the solution of anti-CD16/32 monoclonal antibody (Biolegend, cat. no. 101302, Clone: 93, 1:200 dilution) for blocking FcR-mediated and non-specific antibody binding. Then, the suspension was incubated for 10 min at 25 °C and washed 5 times using FACS buffer. Subsequently, H-2L$^d$ MuLV gp70 tetramer-SPSYVYHQF-PE (MBL International, cat. no. TS-M521-1, 1:50 dilution) was added into the samples and incubated for 20 min at room temperature. Next, fluorescence-labeled antibodies, CD3 (Biolegend, cat. no. 100204, Clone: 17A2, 1:200 dilution) and CD8 (Biolegend, cat. no. 100712, Clone: 53-6.7, 1:200 dilution) were added and incubated for 20 min on ice. Afterwards, the samples were washed twice using FACS buffer. Stained cells were monitored using flow cytometer (BD FACSCalibur) and evaluated using FlowJo software. Gating set up was based on FMO control.

## Immunofluorescence staining

Tumor tissues were harvested and snap-frozen with optimum cutting temperature medium. The tumor tissue was then cut with cryotome, mounted on a slide, and stained using primary antibodies F4/80 (Abcam, cat. no. ab100790, 1:200 dilution) and CD8 (Abcam, cat. no. ab22378, 1:200 dilution) for 12 h at 4 °C. After adding fluorescently labeled secondary antibodies (goat anti-rabbit IgG (H + L); Thermo-Fisher, cat. no. A32733, 1:1000 dilution) and goat anti-rat IgG (H + L; ThermoFisher, cat. no. A18866, 1:600 dilution)), the slide was observed using CLSM (C2SI, Nikon, Japan).

## Statistical analysis

All results are expressed as mean value ± standard deviation. The variations among different groups were assessed via one-way analysis of variance (ANOVA) and Student's $t$ test (two-tailed), and the log-rank test was performed to judge the survival benefit. The exact $P$ value is provided in the corresponding figure. All statistical analyses were

performed with GraphPad Prism 8.0. $P < 0.05$ indicated statistically significant.

## Reporting summary
Further information on research design is available in the Nature Portfolio Reporting Summary linked to this article.

## Data availability
All data supporting the findings of this study are available within the Article, Supplementary Information or Source Data file. The source data underlying Figs. 2d–f, h, i, k, m, o, p, 4a–c, e–g, 5b, c, e–g, 6b–h, 7c–h, Supplementary Figs. 4–6, 8, 9, 12b, c, 13, 15a, 17b, 21, 23–25a, b, 25d–f, 27, 28, 35, 37c–f, 38, and 39 are provided with this paper and are also available in the Figshare database at: https://doi.org/10.6084/m9.figshare.22715356[70]. Source data are provided with this paper.

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

## Acknowledgements

This research was supported by the National Key R&D Program of China (2021YFA0909900 (J.S.) and 2022YFE0111600 (J.S.)), National Natural Science Foundation of China (No. 82073777 (J.S.), 82273874 (J.S.), and 62005284 (Y.L.)), the National University of Singapore Startup Grant (NUHSRO/2020/133/Startup/08 (X.C.)), NUS School of Medicine Nanomedicine Translational Research Program (NUHSRO/2021/034/TRP/09/Nanomedicine (X.C.)), National Medical Research Council (NMRC) Center Grant Program (CG21APR1005 (X.C.)), and Singapore Ministry of Education, Academic Research Fund Tier 2 (T2EP30122-0002 (X.C.)). K.W. acknowledges the China Scholarship Council (CSC) for financially supporting his work at the National University of Singapore (No. 202208210238 (K.W.)).

## Author contributions

All authors have given approval to the final version of the manuscript. K.W., Y.Z., D.W., J.S. and X.C. conceived project and designed experiments; K.W., Y.L., X.W., Z.Z., L.C., X.F., B.W. and F.L. performed research; K.W., Y.L. and X.W. analyzed data; X.Z. and Z.H. provided useful suggestions; K.W., J.S., and X.C. wrote the initial draft. All authors contributed to the writing of the final manuscript.

## Competing interests

The authors declare no competing interests.

## Additional information

[1]Department of Pharmaceutics, Wuya College of Innovation, Shenyang Pharmaceutical University, 103 Wenhua Road, Shenyang, Liaoning 110016, P. R. China. [2]Departments of Diagnostic Radiology, Surgery, Chemical and Biomolecular Engineering, and Biomedical Engineering, Yong Loo Lin School of Medicine and Faculty of Engineering, National University of Singapore, Singapore 119074, Singapore. [3]Key Laboratory of Design and Assembly of Functional Nanostructures, Fujian Institute of Research on the Structure of Matter, Chinese Academy of Sciences, Fuzhou 350002, China. [4]Department of Translational Medicine & Xiamen Key Laboratory of Rare Earth Photoelectric Functional Materials, Xiamen Institute of Rare-Earth Materials, Haixi Institute, Chinese Academy of Sciences, Xiamen 361021, P. R. China. [5]School of Pharmacy, Shenyang Pharmaceutical University, Shenyang, Liaoning 110016, China. [6]Center for AIE Research, College of Materials Science and Engineering, Shenzhen University, Shenzhen 518060, China. [7]National Engineering Research Center for Marine Aquaculture, Marine Science and Technology College, Zhejiang Ocean University, Zhoushan, Zhejiang Province 316004, China. [8]Clinical Imaging Research Centre, Centre for Translational Medicine, Yong Loo Lin School of Medicine, National University of Singapore, Singapore 117599, Singapore. [9]Nanomedicine Translational Research Program, NUS Center for Nanomedicine, Yong Loo Lin School of Medicine, National University of Singapore, Singapore 117597, Singapore. [10]Institute of Molecular and Cell Biology, Agency for Science, Technology, and Research (A*STAR), 61 Biopolis Drive, Proteos, Singapore 138673, Singapore. [11]These authors contributed equally: Kaiyuan Wang, Yang Li, Xia Wang. ✉e-mail: zhouyingtang@zjou.edu.cn; wangd@szu.edu.cn; sunjin@syphu.edu.cn; chen.shawn@nus.edu.sg

