## [Peer Review File · Nature Communications]

Gas therapy potentiates aggregation-induced emission
luminogen-based photoimmunotherapy of poorly
immunogenic tumor through cGAS-STING pathway activationREVIEWER COMMENTS

Reviewer #1 (Remarks to the Author): with expertise in AIEgens, cancer nanotherapy

Wang and coworkers reported the immunoadjuvant effect of tumour microenvironment-driven gas therapy, and introduced gas nanoadjuvant into phototherapy to combat poorly immunogenic tumour. The produced H₂S/CO could exert therapeutic effect by decreasing mitochondrial membrane potential, which would further damage mitochondria, and release mitochondrial DNA into cytoplasm. The exposure of cytoplasmic DNA stimulated immunity through cGAS-STING pathway activation. The gas nanoadjuvant-assisted photoimmunotherapy effectively inhibited both primary tumour and distant tumour in the murine 4T1 TNBC model, and significantly eliminated tumour metastasis and rechallenge. Overall, this manuscript is well written and the overall quality is good. Nevertheless, there are some parts in need of completion and clarification. I have detailed those more carefully below. I recommend publication in Nature Communications after these changes have been incorporated.

Comment 1. The authors should conduct transmission electron microscopy (TEM) measurements to display more nanoparticles in the field of vision.

Comment 2. The intracellular GSH consumption by tvHMS should be evaluated by GSH probe such as ThiolTracker Violet.

Comment 3. The authors need to confirm the ROS level in tumour tissue after in vivo MTHMS+L treatment.

Comment 4. The intratumoural CO level should be determined using FL-CO-1 + PdCl₂ fluorescence probe in mice after MTHMS+L treatment.

Comment 5. The Fig. 1d,i,k, Supplementary Fig.s 7, 8, 11b should be adjusted to show individual values. The authors should provide the calculation methods for the statistical test of all figures containing p value.

Comment 6. It seems that the description of Fig. 4 doesn't follow the order displayed in the figure. The authors should carefully check it and make corresponding modification in the "RESULTS AND DISCUSSION" part.

Comment 7. Please provide scale bars for all the microscopic images, for example, the PA images in Supplementary Fig. 10.

Comment 8. The legends of some figures (e.g., Fig. 5) are too small to be seen clearly. The authors are suggested to enlarge the words.

Comment 9. Please provide the experimental details in the figure captions, e.g., the concentrations of all spectra measurements, photothermal effect, etc. Details need to be provided about how ELISA analysis was performed. More experiment details about the NIR fluorescence and PA imaging should be provided, e.g., the excitation light source, signal collection, and software.

Comment 10. The expression needs further check. I advise authors to standardize the use of technical term for avoiding misunderstanding to readers. For instance, whether "mean hydrodynamic diameter" and "size" mean the same meaning.

Reviewer #2 (Remarks to the Author): with expertise in cancer nanotherapy

This study revealed the immunoadjuvant property of tumor microenvironment-driven gas therapy to activate cGAS-STING signaling for augmented AIEgen-based photoimmunotherapy against poorly immunogenic TNBC. The GSH/NIR sequentially initiated gas nanoadjuvant was rationally constructed through encapsulating AIEgen and MnCO into tetrasulfide-doped virus-mimicking hollow mesoporous silica. Under intratumoral GSH and in situ NIR sequential stimulation, the gas nanoadjuvant achieved tumor-specific amplified H₂S/CO/Mn²⁺ release to trigger immune responses by cGAS-STING pathway activation. The authors presented a novel strategy to assist AIEgen-mediated phototherapy for poorly immunogenic TNBC treatment. This study is innovative, the hypothesis is validated both in vitro and in vivo, and the conclusion is well supported by the experimental data. The relevant discussions and perspectives are profound and significant. The following aspects should be addressed to further promote the work.

1. The in vitro stability of MTHMS should be examined by dynamic light scattering (DLS).
2. The authors need to illustrate the preparation mechanism of tvHMS. For example, how did tetrasulfide bond participate in the formation of tvHMS?
3. For the fabrication of tvHMS, why did NaOH selectively etch the inner solid silica? The underlying mechanism for NaOH etching method to make the virus-like silica nanoparticles should be discussed in depth.
4. The excretion possibility should be discussed to guarantee the potential biocompatibility of nanocarrier.
5. The in-depth mechanism for higher cellular uptake efficiency of virus-like MTHMS than sphere-like MTHMS should be illustrated clearly.
6. There are many self-citations, the broader context for this work should be given.
7. The extensive studies have developed different kinds of nanomaterial-based combination of phototherapy and gas therapy. The authors need to further provide related discussions.
8. The authors need to pay attention to some details. The full name of compounds should be indicated in the first appearance, such as TEOS and CTAB.
9. For AIEgen-based photoimmunotherapy, some important ref. should be cited, such as J. Mater. Chem. C, 2020,8, 15622-15625; Inorganic Chemistry, 2022, doi:10.1021/acs.inorgchem.2c01206.

Reviewer #3 (Remarks to the Author): with expertise in cancer immunology/therapy

This is an interesting interdisciplinary manuscript dealing with the development of multifunctional phototheranostic nanoparticles with additional gas-mediated (H₂S, CO) immunoadjuvant property. The method enables laser-induced intracellular/intratumoral gas release, which leads to release of mitochondrial DNA into the cytosol and thereby activation of the cGAS/STING/Type I IFN pathway. This leads to a potentially improved induction of tumor-specific CD8⁺ T cells through improved tumor antigen cross-presentation by dendritic cells. My criticism relates mainly to the question to what extent the observed antitumor effects actually depend on the immune system.

Major points

1.

On p.11, line 230-234, the authors state: "the frequency of mature DCs induced by tumor cells with different treatments was evaluated as a consequence of cGAS/STING activation. ... As expected, MTHMS+L treated cancer cells significantly boosted DC maturation (Fig. 3d, e), which was driven by the striking IFN- β release after cGAS-STING activation".

Fig. 3d, e suggest some upregulation of CD80/CD86 on BM-derived DCs, but these experiments do not show that these changes are due to IFN β release after cGAS-STING activation. Such a conclusion would require experiments with inhibitors of the cGAS-STING pathway and/or DCs from mutant mice with deficiencies in the cGAS/STING pathway.

2.

Do the antitumor effects observed (the slower growth of the orthotopic tumors, the reduced lung metastasis, the slower growth of the re-challenge tumors, the longer survival of the mice) following MTHMS+L in Fig. 4 depend on the immune system (on CD8 T cells)? This should be answered with T cell depletion experiments.

What was the effect of MTHMS on tumor growth (Fig. 4b-d) and lung metastasis (Fig. 4h) without laser illumination?

3.

p.13. The authors state that they did not find evidence for toxicity in normal organs. What about the region/tissue around the orthotopic tumor? Was there toxicity observed (e.g., edema due to necrosis induced in the tumor?). If so, could this affect the take rate of re-challenge tumors?

4.

Fig.5h, p.15, line 337-340: the authors present an increase in effector memory CD62L^{low} CD44⁺ cells in the spleen from 16% in PBS-treated mice to 47.5% in MTHMS+L-treated mice, arguing that this result indicates durable immunological memory.

I have doubts whether such a great increase in the proportion of these cells in the spleen reflects tumor-specific memory. To draw conclusions as to tumor-specific memory or the induction of tumor-specific T cell immunity, tumor-specific T cells need to be measured (e.g., with MHC tetramers or peptide restimulation).

The same applies to Fig. 6g, h (abscopal model).

5.

In the abscopal model, the secondary tumor was implanted into the left mammary fat pad which is not so far away from the right fat pad.

Is it possible that reduced growth of the secondary tumor is affected by local inflammatory responses due to tissue destruction (e.g., edema following tissue necrosis)?

Is the tumor growth affected by CD8⁺ T cells? This should be assessed by doing these experiments in mice depleted of CD8⁺ T cells by injecting depleting antibodies.

6. Fig. 6. In the abscopal experiments, the nanoparticles were injected one day after implantation of the secondary tumor. Is it possible that the slow growth of the secondary tumor is due to direct effects of the injected nanoparticles? If so, this would not be an abscopal effect. This should be ruled out by control experiments with MTHMS without laser.

7. p.11, lines 251- p.12, line 257:

The time point of the apoptosis measurements seems not to be indicated. This should be done. In addition, a time course for the cell death should be provided.

8.

FMO, or isotope controls should be included for all FC examples, especially for the samples with no clear separation (e.g. Fig. 5e-g) to see how this populations were gated. Authors should provide a gating strategy to also show how other populations were gated (e.g. CD11c in Fig. 3d, and Fig. 5c).

Minor points

1.

The manuscript contains many abbreviations, some of which seem to lack explanation. The readability could be improved by introducing each abbreviation where it comes up first.

Examples: P.6, line 118: TSSI seems not to be defined/explained. Likewise, TEOS, CTAB (p.6); DCFH-DA (p.8); WSP-1 (p.9).

Furthermore, it would be good to explain the full names of the different particle types at a certain place (e.g., a list of abbreviations or a suppl. table) or below each figure (e.g. MHMS, THMS, MTHMS, etc.).

2.

p.12, line 277 – p.13, line 279: the authors describe tumor treatment experiments in mice (Fig. 4), saying that on d10, 4T1 tumor cells were injected i.v. to mimic metastasis.

4T1 is spontaneously metastasizing to the lung. It would be good to explain in the main text why 4T1 tumor cells were additionally injected i.v.

In the main text, it should also be described that the treatment of the orthotopic tumor was done twice before the i.v. inoculation of the tumor cells.

Where and with how many tumor cells were the mice re-challenged?

3.

p.16, line 360: the authors also measured IL-6 in the tumor lysates and argue that this demonstrates reversal of immunosuppression. However, IL-6 usually is associated with protumoral effects.

4.

Suppl. Methods line 236. The quantity of the samples loaded for SDS PAGE should be indicated.

5.

Suppl. Methods line 246: flow cytometry analysis of tumor single-cell suspensions. It should be indicated with which enzyme(s) the tumors were digested.

6. In Fig. 5d and Fig. 6e need to be indicated from which population % of CD8+ T cells is shown. In Fig. 5d, f, and Fig. 6e, the x-axis should be labeled.

Reviewer #4 (Remarks to the Author): with expertise in cancer nanotherapy

In this manuscript, the authors propose the novel viewpoint of gas nanoadjuvant with robust cGAS-STING pathway activation to assist photoimmunotherapy of poorly immunogenic tumor. In this paper, the virus-like surface helps gas nanoadjuvant invade cancer cells more effectively through spike surface-assisted adhesion. And intratumoral overexpressed GSH could break the tetra-sulfide bond to release H₂S. Following NIR laser irradiation, the AIEgen-based phototherapy in situ activated MnCO to generate Mn²⁺ and CO. Both H₂S and CO induced the intracellular release of mtDNA, performing the immunoadjuvant property of engaging cGAS-STING pathway. Furthermore, Mn²⁺ sensitize cGAS to enhance STING-mediated type I IFN response in both tumor cells and DCs. This paper provides a new direction for breaking the bottleneck of poorly immunogenic tumor treatment. Overall, the idea of this paper is interesting and the conclusion is supported by data provided. The following concerns are suggested for improving the manuscript:

Question 1: Photo images of bubble generation of CO gas upon NIR laser triggering in MTHMS with or without H₂O₂ should be added to visualize the phototherapy-facilitated CO generation.

Question 2: In Supplementary Fig. 10a, the NIR-II fluorescence images at different time points after intravenous administration of free TSSI solution should be added as a control.

Question 3: The IC₅₀ values of different treatments in Fig. 3f should be listed in Supplementary and additional material as Supplementary Table.

Question 4: In Supplementary Fig. 8, the variation between sphere-like MTHMS group and virus-like MTHMS group in flow cytometric quantification of cellular uptake should be assessed through Student's t-test.

Question 5: In Supplementary Fig. 14, the legends should be noted in figure to point out the specific treatment of groups.

Question 6: There are many methods to detect CO, authors used hemoglobin here. Authors should provide some references to support their choice in this aspect.

Question 7: In the experimental section of in vitro ROS generation assessment, authors indicated that "the fluorescence was detected through PL instrument at 525 nm". Authors should add the excitation wavelength for the measurement.

Question 8: The detailed statistical analysis methods should be individually indicated in every figure captions.

Question 9: Authors should add the introduction and discussion about the current strategies of cGAS-STING pathway activation in the field of nanomedicine.

Response to Reviewers' comments:

Reviewer #1:

Wang and coworkers reported the immunoadjuvant effect of tumour microenvironment-driven gas therapy, and introduced gas nanoadjuvant into phototherapy to combat poorly immunogenic tumour. The produced H₂S/CO could exert therapeutic effect by decreasing mitochondrial membrane potential, which would further damage mitochondria, and release mitochondrial DNA into cytoplasm. The exposure of cytoplasmic DNA stimulated immunity through cGAS-STING pathway activation. The gas nanoadjuvant-assisted photoimmunotherapy effectively inhibited both primary tumour and distant tumour in the murine 4T1 TNBC model, and significantly eliminated tumour metastasis and rechallenge. Overall, this manuscript is well written and the overall quality is good. Nevertheless, there are some parts in need of completion and clarification. I have detailed those more carefully below. I recommend publication in Nature Communications after these changes have been incorporated.

Comment 1. The authors should conduct transmission electron microscopy (TEM) measurements to display more nanoparticles in the field of vision.

Response: We appreciate the reviewer's comments. As the reviewer suggested, we have conducted transmission electron microscopy (TEM) measurements to display more nanoparticles in the field of vision.

Comment 2. The intracellular GSH consumption by tvHMS should be evaluated by GSH probe such as ThiolTracker Violet.

Response: We appreciate the reviewer's comments. As suggested, we evaluated the intracellular GSH consumption by tvHMS using ThiolTracker Violet. These results were shown in the revised manuscript: **“As seen in ThiolTracker Violet fluorescence images (Supplementary Fig. 10), tvHMS-incubated 4T1 cells showed attenuated green fluorescence due to GSH depletion.”**

Supplementary Fig. 10. ThiolTracker Violet (GSH) fluorescence imaging of 4T1 cancer cells after tvHMS treatment (scale bar = 15 μm).

Comment 3. The authors need to confirm the ROS level in tumour tissue after *in vivo* MTHMS+L treatment.

Response: Thank the reviewer for pointing out this issue. As reviewer suggested, we investigated the ROS level in tumor tissue after *in vivo* MTHMS+L treatment through DCFH-DA staining. These results were discussed in the revised manuscript: “To evaluate the ROS level after MTHMS+L treatment, the DCFH-DA staining of tumor slides was performed. As seen in Supplementary Fig. 19, MTHMS+L treatment induced strong DCF fluorescence intensity in tumor tissue, implying intratumoral ROS burst.”

Supplementary Fig. 19. DCFH-DA (ROS) staining of tumor from mice treated with MTHMS+L. Scale bar = 100 μm .

Comment 4. The intratumoural CO level should be determined using FL-CO-1 + PdCl₂ fluorescence probe in mice after MTHMS+L treatment.

Response: We appreciate the reviewer's comments. As reviewer suggested, we determined the intratumoral CO level after MTHMS+L treatment using FL-CO-1 fluorescence probe. These results were discussed in the revised manuscript: "We then investigated the intratumoral CO level with FL-CO-1 fluorescence probe. As seen in Supplementary Fig. 20, MTHMS+L treatment provoked considerable CO generation in the tumor site."

Supplementary Fig. 20. FL-CO-1 (CO) staining of tumor from mice treated with MTHMS+L. Scale bar = 100 μm .

Comment 5. The Fig. 1d,i,k, Supplementary Figs 7, 8, 11b should be adjusted to show individual values. The authors should provide the calculation methods for the statistical test of all figures containing p value.

Response: We appreciate the reviewer's comments. As reviewer suggested, we have adjusted the Fig. 1d,i,k, Supplementary Figs. 8, 9, 17b to show individual values. And we have provided the calculation methods for the statistical test of all figures containing p value.

Comment 6. It seems that the description of Fig. 4 doesn't follow the order displayed in the figure. The authors should carefully check it and make corresponding modification in the "RESULTS AND DISCUSSION" part.

Response: We appreciate the reviewer's comments. We have carefully checked the description of Fig. 4 and made corresponding modification in the "RESULTS AND DISCUSSION" part to make it follow the order displayed in the figure.

Comment 7. Please provide scale bars for all the microscopic images, for example, the PA images

in Supplementary Fig. 10.

Response: We appreciate the reviewer's comments. We have provided scale bars for all the microscopic images including the PA images in Supplementary Fig. 18.

Comment 8. The legends of some figures (e.g., Fig. 5) are too small to be seen clearly. The authors are suggested to enlarge the words.

Response: We appreciate the reviewer's comments. As suggested, we have increased the font size for better readability.

Comment 9. Please provide the experimental details in the figure captions, e.g., the concentrations of all spectra measurements, photothermal effect, etc. Details need to be provided about how ELISA analysis was performed. More experiment details about the NIR fluorescence and PA imaging should be provided, e.g., the excitation light source, signal collection, and software.

Response: We appreciate the reviewer's comments. As suggested, we have added the experimental details in the figure captions. The details about the ELISA analysis have been supplemented in the revised manuscript: "The standard or sample was added to each well and incubated for 90 min at 37 °C, then the wells were washed for five times. Next, biotin-labeled detection antibody was added and incubated for 60 min at 37 °C, then the wells were washed five times. Subsequently, streptavidin-conjugated HRP was added and incubated for 30 min at 37 °C, then the wells were washed five times. Afterwards, 3,3',5,5'-tetramethylbenzidine (TMB) was added and incubated for 15 min at 37 °C. Finally, stop solution was added and monitored at 450 nm immediately." Moreover, "NIR-II fluorescence imaging was performed on NIR-OPTICS Series III 900/1700 (China) with the long pass (LP) filter of 1000 nm, and PSViewer software was employed to analyze the data. Photoacoustic imaging was recorded using the VisualSonics Vevo-2100 imaging system (Canada), and VevoLAZR software (Fujifilm, VisualSonics) was employed to analyze the data."

Comment 10. The expression needs further check. I advise authors to standardize the use of technical term for avoiding misunderstanding to readers. For instance, whether "mean hydrodynamic diameter" and "size" mean the same meaning.

Response: We appreciate the reviewer's comments. We have carefully checked the expression and standardized the use of technical terms to avoiding misunderstanding. "Mean hydrodynamic diameter" and "size" have the same meaning, we have revised "size" to "mean hydrodynamic diameter".

Reviewer #2:

This study revealed the immunoadjuvant property of tumor microenvironment-driven gas therapy to activate cGAS-STING signaling for augmented AIEgen-based photoimmunotherapy against poorly immunogenic TNBC. The GSH/NIR sequentially initiated gas nanoadjuvant was rationally constructed through encapsulating AIEgen and MnCO into tetrasulfide-doped virus-mimicking hollow mesoporous silica. Under intratumoral GSH and in situ NIR sequential stimulation, the gas nanoadjuvant achieved tumor-specific amplified H₂S/CO/Mn²⁺ release to trigger immune responses by cGAS-STING pathway activation. The authors presented a novel strategy to assist AIEgen-mediated phototherapy for poorly immunogenic TNBC treatment. This study is innovative, the hypothesis is validated both *in vitro* and *in vivo*, and the conclusion is well supported by the experimental data. The relevant discussions and perspectives are profound and significant. The following aspects should be addressed to further promote the work.

1. The *in vitro* stability of MTHMS should be examined by dynamic light scattering (DLS).

Response: We appreciate the reviewer's comments. As suggested, we examined the *in vitro* stability of MTHMS by DLS. "The solution of MTHMS was then incubated in PBS or PBS + 10 % FBS. At designated time, aliquot of the solution was taken and monitored through DLS. As shown in Supplementary Fig. 6, the mean diameter of MTHMS in PBS or 10 % FBS solution displayed negligible change after storage at ambient condition for 24 h, suggesting good *in vitro* stability of the MTHMS."

Supplementary Fig. 6. The stability of MTHMS in PBS or PBS + 10 % FBS.

2. The authors need to illustrate the preparation mechanism of tvHMS. For example, how did tetrasulfide bond participate in the formation of tvHMS?

Response: We appreciate the reviewer's comments. It was reported that owing to the chemical homology between tetraethyl orthosilicate (TEOS) and bis(triethoxysilyl propyl) disulfide (BTESPD) ("chemical homology" principle, *Adv Mater.* 2015, 27, 215-222), BTESPD can be used to co-hydrolyze and co-condensate with TEOS to form an organic-inorganic hybrid silica nanosystem on a molecular level (*Biomaterials.* 2018, 161, 292-305). Moreover, TEOS as a frequently-used silicon source could be co-hydrolyzed and co-condensated into the "functional bond"-doped (e.g., tetrasulfide(-S-S-S-S)-doped; diselenide(-Se-Se-)-doped) organic-inorganic hybrid silica nanosystem with "functional bond"-doped TEOS analogue (e.g., bis[3-(triethoxysilyl)propyl] tetrasulfide (TESPT, tetrasulfide-bridged silicon source: *Adv Mater.* 2021, 33, 2101223; *Nat Commun.* 2019, 10, 1241; *Biomaterials.* 2021, 277, 121074; *Theranostics.* 2020, 10, 2918-2929; *Microporous and Mesoporous Materials.* 2020, 302, 110228); bis[3-(triethoxysilyl)propyl] diselenide (BTESePD, diselenide-bridged silicon source: *Adv Mater.* 2018, 30, 1801198)) (Fig. R1).

Figure R1. Chemical structures of TEOS and TESPT. The two chemical molecules possess chemical homology.

To improve the readability, The preparation mechanism of tvHMS has been supplemented in the revised manuscript: **“The tvHMS was fabricated *via* post-co-condensation of TESPT with TEOS (v/v, 1: 3) through the chemical homology principle on the solid silica surface.”**

3. For the fabrication of tvHMS, why did NaOH selectively etch the inner solid silica? The underlying mechanism for NaOH etching method to make the virus-like silica nanoparticles should be discussed in depth.

Response: We appreciate the reviewer's comments. We suspect that the formation of surface morphology was likely due to the dosage and etching time of NaOH. In addition, the concentration and purity (> 99%, Sigma-Aldrich) of CTAB should not be ignored.

“We listed the underlying mechanism of the conversion of core-shell structure to the virus-mimicking hollow mesoporous structure by NaOH etching: (1) NaOH rapidly etched the solid silica to produce dissolved silicate species, the core was removed to yield the hollow structure; (2) NaOH

slowly etched a part of mesoporous silica to produce dissolved silicate species, leaving behind the disordered, pot-holed surface layer while generating the spike structure.” The above detailed mechanism has been added to the revised manuscript.

4. The excretion possibility should be discussed to guarantee the potential biocompatibility of nanocarrier.

Response: We appreciate the reviewer’s comments. The hollow mesoporous silica nanoparticles have attracted the interest of the scientific community due to their potential to be applied in the nanomedicine field (*Biomaterials*. 2010, 31, 5564-5574; *Angewandte Chemie*. 2016, 128(5): 1931-1935). The main advantages of hollow mesoporous silica nanoparticles arise from their simple, scalable, and cost-effective fabrication as well as their non-toxic matrix structure, large pore volume and surface area that is prone to be functionalized. A critical factor that affects nanocarrier application in biomedical applications is their biocompatibility. As a promising candidate material for efficient drug delivery, the biodegradation and excretion of mesoporous silica were characterized in many studies (*Small*. 2010, 6, 1794-1805, *Microporous and Mesoporous Materials*. 2016, 236, 141-157). Silica is an endogenous substance of the human body that is particularly abundant in supporting tissues (*Nanomedicine*. 2012, 7, 111-120). Mesoporous silica is a biodegradable material and can be degraded by simulated body fluids within 15 days in vitro (*Microporous and Mesoporous Materials*. 2010, 131, 314-320). Moreover, the silica nanoparticles in the form of Cornell dots, which assist both the diagnosis and targeted treatment of cancer cells, have been tested in human (*Sci Transl Med*. 2014, 6, 260ra149). The results showed no toxic or adverse events attributable to the silica particles, highlighting the great potential of silica nanoparticles in clinical applications.

As reviewer suggested, the excretion possibility was discussed in the revised manuscript: “Moreover, mesoporous silica has been widely applied in biomedicine and exhibited great biocompatibility. It was demonstrated that mesoporous silica is excreted through feces and urine. It was reported that urinary excretion could account for 15-45% of the injected mesoporous silica nanoparticles at 0.5 h post-administration (*Small*. 2011, 7, 271-280). The excellent biodegradability guarantees safety for clinical application.”

5. The in-depth mechanism for higher cellular uptake efficiency of virus-like MTHMS than sphere-like MTHMS should be illustrated clearly.

Response: We appreciate the reviewer’s comments. The surface topography and structure of nanoparticles have been demonstrated to significantly influence the cellular uptake process (*Nat Mater*. 2009, 8, 543-557). Yu et al., Xu et al., and Yu et al., respectively, reported that rambutan-like

and rattle-like nanoparticles exhibited enhanced cellular internalization compared with spherical counterparts, revealing the rough surface-enhanced adhesion (*J Am Chem Soc.* 2016, 138, 6455-6462; *ACS Nano.* 2018, 12, 5646-5656; *Adv Mater.* 2013, 25, 6233-6237). Besides, Zhang et al. reported a work, titled “Why synthetic virus-like nanoparticles can achieve higher cellular uptake efficiency?” (*Nanoscale.* 2020, 12, 14911-14918), which further supported our study. Therefore, we believe that the in-depth mechanism behind the higher cell uptake of virus-like MTHMS than sphere-like MTHMS is the virus-like nanoparticles exhibited much more contacting sites per unit area with cell membranes compared to sphere-like nanoparticles, which was beneficial to elevate the adhesion interaction to significantly enhance the cell entry efficiency.

Following the reviewer’s suggestion, the related discussion and references have been added in the revised manuscript: “Compared to the smooth surface of sphere-like MTHMS, the rough surface endowed virus-like MTHMS with much more contacting chances and elevated adhesion interaction with cytomembrane to remarkably improve the cellular internalization efficacy.”

6. There are many self-citations, the broader context for this work should be given.

Response: We appreciate the reviewer’s comments. The related references were added to support the background. The relevant articles were added to the revised manuscript: *J Mater Chem C.* 2020, 8, 15622-15625; *Inorg Chem.* 2023, 62, 1786-1790; *Angewandte Chemie.* 2022, 61, e202117798; *Chem Soc Rev.* 2021, 50, 5086-5125; *Biomaterials.* 2018, 185, 51-62. In addition, some self-cited articles were removed from the manuscript.

7. The extensive studies have developed different kinds of nanomaterial-based combination of phototherapy and gas therapy. The authors need to further provide related discussions.

Response: We appreciate the reviewer’s comments. According to the reviewer’s suggestion, we added the related discussions about nanomaterial-based combination of phototherapy and gas therapy in the revised manuscript: “In recent years, diverse efforts have been devoted to eliminating tumors through the nanomaterial-based combination of phototherapy and gas therapy (*Chem Soc Rev.* 2021, 50, 5086-5125; *Chem Eng J.* 2022, 444, 136512). For example, Wan and coworkers loaded biocompatible L-arginine into PCN-224 as an NO donor to combine PDT and NO gas therapy (*Biomaterials.* 2018, 185, 51-62). Upon laser exposure, the donor L-arginine could react with ROS and H₂O₂ to produce NO with a wide diffusion range and long half-life. In the hypoxic microenvironment, NO could sensitize cancer cells to PDT-generated ROS and almost completely eradicate cancer. However, the immunostimulating property of gas therapy was rarely explored to assist photoimmunotherapy.”

8. The authors need to pay attention to some details. The full name of compounds should be indicated in the first appearance, such as TEOS and CTAB.

Response: We appreciate the reviewer's comments. As suggested, we have carefully checked and introduced each abbreviation where it comes up first. Furthermore, we have explained the full names of the different particle types in Supplementary Table 1.

Supplementary Table 1. The full names of different formulations.

Formulations	Full names
dvHMS	disulfide bond incorporated virus-like hollow mesoporous silica
tsHMS	tetrasulfide-functionalized sphere-like hollow mesoporous silica
tvHMS	tetrasulfide-functionalized virus-like hollow mesoporous silica
MHMS	manganese carbonyl (MnCO) encapsulated tvHMS
THMS	TSSI encapsulated tvHMS
MTHMS	MnCO and TSSI co-encapsulated tvHMS

9. For AIEgen-based photoimmunotherapy, some important ref. should be cited, such as J. Mater. Chem. C, 2020,8, 15622-15625; Inorganic Chemistry, 2022, doi:10.1021/acs.inorgchem.2c01206.

Response: We appreciate the reviewer's comments. As suggested, we further cited some important ref. about AIEgen-based photoimmunotherapy in the revised manuscript: *J Mater Chem C*. 2020, 8, 15622-15625; *Inorg Chem*. 2023, 62, 1786-1790; *Angewandte Chemie*. 2022, 61, e202117798.

Reviewer #3:

This is an interesting interdisciplinary manuscript dealing with the development of multifunctional phototheranostic nanoparticles with additional gas-mediated (H₂S, CO) immunoadjuvant property. The method enables laser-induced intracellular/intratatumoral gas release, which leads to release of mitochondrial DNA into the cytosol and thereby activation of the cGAS/STING/Type I IFN pathway. This leads to a potentially improved induction of tumor-specific CD8⁺ T cells through improved tumor antigen cross-presentation by dendritic cells. My criticism relates mainly to the question to what extent the observed antitumor effects actually depend on the immune system.

Major points

1. On p.11, line 230-234, the authors state: “the frequency of mature DCs induced by tumor cells with different treatments was evaluated as a consequence of cGAS/STING activation. ... As expected, MTHMS+L treated cancer cells significantly boosted DC maturation (Fig. 3d, e), which was driven by the striking IFN- β release after cGAS-STING activation”.

Fig. 3d, e suggest some upregulation of CD80/CD86 on BM-derived DCs, but these experiments do not show that these changes are due to IFN β release after cGAS-STING activation. Such a conclusion would require experiments with inhibitors of the cGAS-STING pathway and/or DCs from mutant mice with deficiencies in the cGAS/STING pathway.

Response: We appreciate the reviewer’s comments. To confirm the cGAS-STING activation-induced DC maturation, the highly potent and selective inhibitors (RU.521, C-178, or C-176) of cGAS-STING pathway were separately added when incubating BMDCs with MTHMS+L treated cancer cells (*Nature*. 2018, 559, 269-273; *Nat Commun*. 2017, 8, 750; *MedChemComm*. 2019, 10, 1999-2023). “As shown in Supplementary Fig. 12, the cGAS-STING pathway inhibition significantly decreased the IFN- β , CXCL10, and IL-6 expression and remarkably reduced the frequency of mature DCs, evidencing the cGAS-STING pathway-dependent DC maturation.”

Supplementary Fig. 12. (a) Flow cytometric assessment images and (b) relative quantification of DC maturation (CD11c⁺CD80⁺CD86⁺) triggered by MTHMS+L treated cancer cells with or without cGAS-STING inhibitors (RU.521, C-178, or C-176) (n = 3). (c) The detection of cytokines (IFN- β , CXCL10, and IL-6) in culture supernatants of BMDCs incubated with MTHMS+L treated cancer cells with or without RU.521, C-178, or C-176 (n = 3). Data represent the mean \pm s.d. Statistical significance was calculated through one-way ANOVA using a Tukey post-hoc test.

2. Do the antitumor effects observed (the slower growth of the orthotopic tumors, the reduced

lung metastasis, the slower growth of the re-challenge tumors, the longer survival of the mice) following MTHMS+L in Fig. 4 depend on the immune system (on CD8 T cells)? This should be answered with T cell depletion experiments.

What was the effect of MTHMS on tumor growth (Fig. 4b-d) and lung metastasis (Fig. 4h) without laser illumination?

Response: We appreciate the reviewer's comments. To investigate whether the observed antitumor effects depend on the immune system (on CD8 T cells), 100 µg anti-CD8a antibody (BioXCell, clone 2.43) was intraperitoneally injected into the mice every four days since day 0. "The results showed that CD8⁺ T cell depletion significantly impaired tumor suppression, led to severe lung metastasis, and shortened the survival time of mice treated with MTHMS+L (Supplementary Fig. 25a-e, 26, 27), confirming the central role of the immune system in gas nanoadjuvant-assisted photoimmunotherapy." Meanwhile, the survived mice in MTHMS+L group were also rechallenged with 4T1 cells with/without CD8⁺ T cell depletion (intraperitoneal injection of 100 µg anti-CD8a antibody every four days since tumor rechallenge). "As shown in Supplementary Fig. 25f, the tumor volume of CD8⁺ T cell-depleted mice exhibited an evident increase and reached ~780 mm³ on day 20, validating the effect of the immune system in resistance to tumor relapse."

We also evaluated the effect of MTHMS on tumor growth and lung metastasis without laser illumination. The lack of laser irradiation resulted in the failure of PDT/PTT therapy and insufficient MnCO activation, which remarkably hindered the sequentially initiated stimulation of gas nanoadjuvant. "As shown in Supplementary Fig. 25a-e, 26, 27, MTHMS treatment displayed a slight delay of tumor progression (~1060 mm³) and a large number of lung metastasis nodules, indicating a significantly attenuated therapeutic effect."

Supplementary Fig. 25. Assessment of therapeutic efficiency of MTHMS (G1: Saline, G2: MTHMS+L+aCD8, G3: MTHMS, G4: MTHMS+L). (a) Tumor growth curve (n = 5), (b) tumor weight variations (n = 5), (c) tumor graphs (n = 5), (d) body weight changes (n = 5) and (e) survival curve (n = 13) following different treatments. (f) Tumor growth curve of mice rechallenged with 4T1 cancer cells (n = 5). Data represent the mean \pm s.d. Statistical significance was calculated through one-way ANOVA using a Tukey post-hoc test (a, b, f) or log-rank (Mantel-Cox) test (e).

Supplementary Fig. 26. (a) H&E, TUNEL, and Ki67 staining of tumor slices collected from mice receiving various treatments. (b) Photos of lung stained with Bouin's fluid and H&E staining of lung and liver following different treatments. Scale bar = 100 μ m.

Supplementary Fig. 27. The average number of surface lung metastases ($n = 5$). Data represent the mean \pm s.d. Statistical significance was calculated through one-way ANOVA using a Tukey post-hoc test.

3. p.13. The authors state that they did not find evidence for toxicity in normal organs. What about the region/tissue around the orthotopic tumor? Was there toxicity observed (e.g., edema due to necrosis induced in the tumor?). If so, could this affect the take rate of re-challenge tumors?

Response: We appreciate the reviewer's comments. As suggested, the tissue around the orthotopic tumor was dissected to observe the toxicity. As shown in Fig. R2, the skin, subcutaneous tissue, and muscle were normal without evidence of edema, hyperplasia, degeneration, hemorrhage, fibrosis, or necrosis. In addition, the proliferation and apoptosis of normal tissue and tumor tissue after MTHMS+L treatment were compared through Ki67 and TUNEL staining. As shown in Fig. R3, the tumor tissue exhibited widespread apoptosis, whereas the normal tissue showed high proliferative activity with negligible apoptosis. Moreover, the resistance against rechallenged tumors depended on the adaptive immunity as indicated by the severe relapse of surviving mice in the MTHMS+L group with CD8⁺ T cell depletion (Supplementary Fig. 25f).

Figure R2. H&E staining of the skin, subcutaneous tissue, and muscle around the orthotopic tumor after MTHMS+L treatment. Scale bar = 100 μ m.

Figure R3. (a) TUNEL and (b) Ki67 staining of tumor tissue (TT) and normal tissue (NT) around the tumor collected from mice receiving MTHMS+L treatment. Scale bar = 100 μ m.

Supplementary Fig. 25. Assessment of therapeutic efficacy of MTHMS (G1: Saline, G2: MTHMS+L+aCD8, G3: MTHMS, G4: MTHMS+L). (a) Tumor growth curve (n = 5), (b) tumor weight variations (n = 5), (c) tumor graphs (n = 5), (d) body weight changes (n = 5) and (e) survival curve (n = 13) following different treatments. (f) Tumor growth curve of mice rechallenged with 4T1 cancer cells (n = 5). Data represent the mean \pm s.d. Statistical significance was calculated through one-way ANOVA using a Tukey post-hoc test (a, b, f) or log-rank (Mantel-Cox) test (e).

4. Fig.5h, p.15, line 337-340: the authors present an increase in effector memory CD62L^{low} CD44⁺ cells in the spleen from 16% in PBS-treated mice to 47.5% in MTHMS+L-treated mice, arguing that this result indicates durable immunological memory.

I have doubts whether such a great increase in the proportion of these cells in the spleen reflects tumor-specific memory. To draw conclusions as to tumor-specific memory or the induction of tumor-specific T cell immunity, tumor-specific T cells need to be measured (e.g., with MHC tetramers or peptide restimulation).

The same applies to Fig. 6g, h (abscopal model).

Response: We appreciate the reviewer's comments. The murine leukemia virus envelope glycoprotein gp70 is present in many mouse tumor cell lines, while gp70 is normally silent in normal mouse tissue. It can function as a tumor neoantigen in the 4T1 cell line (*Nat Commun.* 2022, 13, 5413; *Proc Natl Acad Sci.* 2018, 115, 8179-8184). To assess the tumor-specific T cell responses triggered by gas nanoadjuvant-assisted photoimmunotherapy, the percentage of gp70 tetramer-specific CD8⁺ T cells in the spleen was monitored by flow cytometry analysis. "Following MTHMS+L treatment,

the proportion of tumor-reactive gp70 tetramer-specific CD8⁺ T cells raised to a high level of ~4.5 % (Supplementary Fig. 35), indicating the induction of tumor-specific T cell immunity.”

We also evaluated the activation of tumor-specific T cell responses in the abscopal model. “As shown in Supplementary Fig. 38, the MTHMS+L evoked the peak frequency of ~4.1% gp70 tetramer-specific CD8⁺ T cells, manifesting the elicitation of robust tumor-specific T cell immunity.” Moreover, MTHMS+L treatment resulted in improved levels of proinflammatory cytokines (IFN- γ , TNF- α , and IL-6) in serum (Fig. R4).

Supplementary Fig. 35. Flow cytometric assay and relative quantification of gp70 tetramer staining of CD8⁺ T cells in spleen in 4T1 tumor model (n = 4). Data represent the mean \pm s.d. Statistical significance was calculated through one-way ANOVA using a Tukey post-hoc test.

Supplementary Fig. 38. Flow cytometric assay and relative quantification of gp70 tetramer staining of CD8⁺ T cells in spleen in the bilateral tumor model (n = 4). Data represent the mean \pm s.d. Statistical significance was calculated through two-tailed student’s t-test.

Figure R4. The secretion of cytokines (IL-6, TNF- α , and IFN- γ) in serum. Data represent the mean \pm s.d. Statistical significance was calculated through two-tailed student's t-test.

5. In the abscopal model, the secondary tumor was implanted into the left mammary fat pad which is not so far away from the right fat pad.

Is it possible that reduced growth of the secondary tumor is affected by local inflammatory responses due to tissue destruction (e.g., edema following tissue necrosis)?

Is the tumor growth affected by CD8⁺ T cells? This should be assessed by doing these experiments in mice depleted of CD8⁺ T cells by injecting depleting antibodies.

Response: We appreciate the reviewer's comments. The region/tissue around the orthotopic tumor after MTHMS+L treatment was dissected and no evidence of edema, hyperplasia, degeneration, hemorrhage, fibrosis, or necrosis was observed (Fig. R2). Additionally, the comparison of proliferation and apoptosis of normal tissue and tumor tissue after MTHMS+L treatment indicated that the tumor tissue exhibited widespread apoptosis, whereas the normal tissue showed high proliferative activity with negligible apoptosis (Fig. R3). To explore whether the tumor growth was affected by CD8⁺ T cells, 100 μ g anti-CD8a antibody (BioXCell, clone 2.43) was intraperitoneally injected into the mice every four days since day 7. **“The results showed that CD8⁺ T cell depletion remarkably impaired the suppression of the secondary tumor after MTHMS+L treatment (Supplementary Fig. 37), verifying the critical role of CD8⁺ T cells in the reduced growth of the secondary tumor.”**

Figure R2. H&E staining of the skin, subcutaneous tissue, and muscle around the orthotopic tumor after MTHMS+L treatment. Scale bar = 100 μ m.

Figure R3. (a) TUNEL and (b) Ki67 staining of tumor tissue (TT) and normal tissue (NT) around the tumor collected from mice receiving MTHMS+L treatment. Scale bar = 100 μ m.

Supplementary Fig. 37. Assessment of therapeutic efficacy of MTHMS in abscopal model (G1: Saline+L, G2: MTHMS+L+aCD8, G3: MTHMS, G4: MTHMS+L). Tumor photos of (a) primary tumor (right) and (b) distant tumor (left). Tumor growth profiles of (c) primary tumor (right) and (d) distant tumor (left). (e) Tumor weight variations and (f) body weight changes after the indicated treatments (n = 5). Data represent the mean \pm s.d. Statistical significance was calculated through one-way ANOVA using a Tukey post-hoc test.

6. Fig. 6. In the abscopal experiments, the nanoparticles were injected one day after implantation of the secondary tumor. Is it possible that the slow growth of the secondary tumor is due to direct effects of the injected nanoparticles? If so, this would not be an abscopal effect. This should be ruled out by control experiments with MTHMS without laser.

Response: We appreciate the reviewer's comments. To rule out the direct effects of the injected nanoparticles in the abscopal experiments, we evaluated the therapeutic outcome of MTHMS without laser irradiation. As shown in Supplementary Fig. 37, the volume of the secondary tumor increased significantly after MTHMS administration without laser irradiation, whereas MTHMS+L treatment efficiently suppressed the secondary tumor, confirming the abscopal effect of gas nanoadjuvant-assisted photoimmunotherapy.

Supplementary Fig. 37. Assessment of therapeutic efficacy of MTHMS in abscopal model (G1: Saline+L, G2: MTHMS+L+aCD8, G3: MTHMS, G4: MTHMS+L). Tumor photos of (a) primary tumor (right) and (b) distant tumor (left). Tumor growth profiles of (c) primary tumor (right) and (d) distant tumor (left). (d) Tumor weight variations and (e) body weight changes after the indicated treatments (n = 5). Data represent the mean ± s.d. Statistical significance was calculated through one-way ANOVA using a Tukey post-hoc test.

7. p.11, lines 251- p.12, line 257: The time point of the apoptosis measurements seems not to be indicated. This should be done. In addition, a time course for the cell death should be provided.

Response: We appreciate the reviewer's comments. For the apoptosis measurements, 4T1 cells inoculated into 6-well plates were processed with: (1) PBS; (2) PBS+L; (3) MHMS; (4) MnCO+TSSI+L; (5) THMS+L; (6) MTHMS+L. After incubation for 12 h, cells in groups (2), (4), (5), and (6) were illuminated with 660 nm laser (0.3 W cm^{-2}) for 5 min. Following further 12 h incubation, cell apoptosis was assessed with Annexin V-FITC/PI staining. As reviewer suggested, the time point of the apoptosis measurement was indicated in the revised manuscript. In addition, the cell apoptosis and live/dead staining of various time points (0 h, 1 h, 3 h, 6 h, 12 h) after laser irradiation were assessed to provide a time course for the cell death following MTHMS+L treatment. As shown in Supplementary Fig. 15a, b, the apoptosis ratio and cell death were gradually increased with prolonged incubation time after laser irradiation.

Supplementary Fig. 15. (a) Flow cytometric assessment images and relative quantification of apoptosis of 4T1 cancer cells. After incubation with MTHMS for 12 h, cells were then illuminated with 660 nm irradiation (0.3 W cm^{-2}) for 5 min. Following further 0, 1, 3, 6, 12 h incubation, cells were stained with Annexin V-FITC/PI ($n = 3$). (b) Calcein-AM/PI staining of 4T1 cells at 0, 1, 3, 6, 12 h after laser irradiation (scale bar = 40 μm).

8. FMO, or isotope controls should be included for all FC examples, especially for the samples with no clear separation (e.g. Fig. 5e-g) to see how this populations were gated. Authors should provide a gating strategy to also show how other populations were gated (e.g. CD11c in Fig. 3d, and Fig. 5c).

Response: We appreciate the reviewer's comments. As suggested, FMO controls were added to all FC examples to see how the populations were gated. We also provided a gating strategy to show how other populations were gated.

Supplementary Fig. 11. Gating strategy and fluorescence minus one (FMO) control for the flow cytometry analysis of DC maturation *in vitro*.

Supplementary Fig. 14. Gating strategy and fluorescence minus one (FMO) control for the flow cytometry analysis of apoptosis of 4T1 cancer cells.

Supplementary Fig. 29. Gating strategy and fluorescence minus one (FMO) control for the flow cytometry analysis of DC maturation in TDLNs.

Supplementary Fig. 30. Gating strategy and fluorescence minus one (FMO) control for the flow cytometry analysis of tumor infiltrating CD8⁺ T cells and CD4⁺Foxp3⁺ Tregs.

Supplementary Fig. 31. Gating strategy and fluorescence minus one (FMO) control for the flow cytometry analysis of M2-like macrophages ($CD206^{hi}CD11b^{+}F4/80^{+}$) in tumor.

Supplementary Fig. 32. Gating strategy and fluorescence minus one (FMO) control for the flow cytometry analysis of M1-like macrophages ($CD80^{hi}CD11b^{+}F4/80^{+}$) in tumor.

Supplementary Fig. 34. Gating strategy and fluorescence minus one (FMO) control for the flow cytometry analysis of $CD3^{+}CD8^{+}CD62L^{low}CD44^{hi}$ T_{EM} in spleen.

Supplementary Fig. 36. Gating strategy and fluorescence minus one (FMO) control for the flow cytometry analysis of gp70 tetramer specific $CD8^{+}$ T cells in spleen.

Minor points

1. The manuscript contains many abbreviations, some of which seem to lack explanation. The readability could be improved by introducing each abbreviation where it comes up first.

Examples: P.6, line 118: TSSI seems not to be defined/explained. Likewise, TEOS, CTAB (p.6); DCFH-DA (p.8); WSP-1 (p.9).

Furthermore, it would be good to explain the full names of the different particle types at a certain place (e.g., a list of abbreviations or a suppl. table) or below each figure (e.g. MHMS, THMS, MTHMS, etc.).

Response: We appreciate the reviewer's comments. As suggested, we carefully checked and introduced each abbreviation where it comes up first. Furthermore, we explained the full names of the different particle types in Supplementary Table 1.

Supplementary Table 1. The full names of different formulations.

Formulations	Full names
dvHMS	disulfide bond incorporated virus-like hollow mesoporous silica
tsHMS	tetrasulfide-functionalized sphere-like hollow mesoporous silica
tvHMS	tetrasulfide-functionalized virus-like hollow mesoporous silica
MHMS	manganese carbonyl (MnCO) encapsulated tvHMS
THMS	TSSI encapsulated tvHMS
MTHMS	MnCO and TSSI co-encapsulated tvHMS

2. p.12, line 277 – p.13, line 279: the authors describe tumor treatment experiments in mice (Fig. 4), saying that on d10, 4T1 tumor cells were injected i.v. to mimic metastasis.

4T1 is spontaneously metastasizing to the lung. It would be good to explain in the main text why 4T1 tumor cells were additionally injected i.v.

In the main text, it should also be described that the treatment of the orthotopic tumor was done twice before the i.v. inoculation of the tumor cells.

Where and with how many tumor cells were the mice re-challenged?

Response: We appreciate the reviewer's comments. As suggested, we explained in the main text why 4T1 tumor cells were additionally injected i.v. "The additional i.v. injection of 4T1 tumor cells into tumor-bearing mice to simulate hematogenous metastasis has been widely applied as an artificial whole-body spreading tumor model (*Nat Commun.* 2019, 10, 2025; *Nat Commun.* 2016, 7, 13193; *ACS Nano.* 2019, 13, 5662-5673; *Nano Today.* 2020, 35, 100987). Compared with spontaneous lung metastasis, the whole-body metastasis model was more aggressive and challenging, which was

suitable for specialized anti-metastasis evaluation.”

According to the reviewer’s suggestion, we described in the main text that “the treatment of the orthotopic tumor was done twice before the i.v. inoculation of the tumor cells”.

For tumor rechallenge study, 2×10^5 4T1 cells were injected into the left mammary fat pad of survived mice.

3. p.16, line 360: the authors also measured IL-6 in the tumor lysates and argue that this demonstrates reversal of immunosuppression. However, IL-6 usually is associated with protumoral effects.

Response: We appreciate the reviewer’s comments. Many studies have revealed that the activation of cGAS-STING pathway would exhibit apparent induction of type I IFNs and some proinflammatory cytokines including IL-6 and TNF- α (*Cell Res.* 2020, 30, 966-979; *Adv Mater.* 2022, 34, 2105783; *ACS Nano.* 2020, 14, 3927-3940). Actually, IL-6 exerts the dual faces in the tumor microenvironment; the dark face that drives malignancy, and the fairer aspect that promotes antitumor adaptive immunity (*Semin Immunol.* 2014, 26, 38-47). Of the proinflammatory cytokines, accumulating evidence establishes IL-6 as a key player in the activation, proliferation, and survival of lymphocytes during active immune responses. IL-6 signaling can also mobilize the T cell immune response, shifting it from a suppressive to a responsive state that can effectively act against tumors. Finally, IL-6 plays an indispensable role in boosting T cell trafficking to lymph nodes and to tumor sites, where they can become activated and execute their cytotoxic effector functions, respectively. To avoid misleading, the manuscript was revised to “MTHMS+L treatment led to high expression of proinflammatory cytokines in serum including TNF- α , IFN- γ , and IL-6 (Fig. 6h).”

4. Suppl. Methods line 236. The quantity of the samples loaded for SDS PAGE should be indicated.

Response: We appreciate the reviewer’s comments. As reviewer suggested, we indicated that equal amount of proteins (20 μ g) were loaded for SDS PAGE.

5. Suppl. Methods line 246: flow cytometry analysis of tumor single-cell suspensions. It should be indicated with which enzyme(s) the tumors were digested.

Response: We appreciate the reviewer’s comments. As reviewer suggested, we indicated that the enzymes used to digest tumors were collagenase IV, hyaluronidase, and deoxyribonuclease I.

6. In Fig. 5d and Fig. 6e need to be indicated from which population % of CD8+ T cells is shown.

In Fig. 5d, f, and Fig. 6e, the x-axis should be labeled.

Response: We appreciate the reviewer's comments. In Fig. 5d and Fig. 6e, we indicated that the population % of CD8⁺ in CD3⁺ T cells is shown. And the x-axis was labeled in Fig. 5d, f, and Fig. 6e.

Reviewer #4:

In this manuscript, the authors propose the novel viewpoint of gas nanoadjuvant with robust cGAS-STING pathway activation to assist photoimmunotherapy of poorly immunogenic tumor. In this paper, the virus-like surface helps gas nanoadjuvant invade cancer cells more effectively through spike surface-assisted adhesion. And intratumoral overexpressed GSH could break the tetra-sulfide bond to release H₂S. Following NIR laser irradiation, the AIEgen-based phototherapy in situ activated MnCO to generate Mn²⁺ and CO. Both H₂S and CO induced the intracellular release of mtDNA, performing the immunoadjuvant property of engaging cGAS-STING pathway. Furthermore, Mn²⁺ sensitize cGAS to enhance STING-mediated type I IFN response in both tumor cells and DCs. This paper provides a new direction for breaking the bottleneck of poorly immunogenic tumor treatment. Overall, the idea of this paper is interesting and the conclusion is supported by data provided. The following concerns are suggested for improving the manuscript:

Question 1: Photo images of bubble generation of CO gas upon NIR laser triggering in MTHMS with or without H₂O₂ should be added to visualize the phototherapy-facilitated CO generation.

Response: We appreciate the reviewer's comments. To confirm the CO generation, the phototherapy-facilitated CO burst from MTHMS upon NIR laser irradiation with or without H₂O₂ was directly visualized by a digital camera. As shown in Fig. R5, a certain number of bubbles from H₂O₂-incubated MTHMS were observed, which was consistent with the previous report that H₂O₂ could trigger MnCO to release CO gas. Moreover, a significantly higher number of bubbles were achieved in the MTHMS solution with both NIR laser and H₂O₂ treatment, due to PDT-triggered ROS generation enhanced oxidant sensitivity of MnCO and the PTT-induced temperature elevation breakdown of the Mn-CO coordination bond to further release extra CO gas.

Figure R5. Photo images of bubble generation of CO gas upon NIR laser irradiation in MTHMS with or without H₂O₂.

Question 2: In Supplementary Fig. 10a, the NIR-II fluorescence images at different time points after intravenous administration of free TSSI solution should be added as a control.

Response: We appreciate the reviewer's comments. According to the reviewer's suggestion, we added the NIR-II fluorescence images at different time points after intravenous administration of free TSSI solution as a control in Supplementary Fig. 16.

Supplementary Fig. 16. NIR-II fluorescence images of tumor-bearing mice at different time points after intravenous administration of free TSSI solution and MTHMS.

Question 3: The IC₅₀ values of different treatments in Fig. 3f should be listed in Supplementary and additional material as Supplementary Table.

Response: We appreciate the reviewer's comments. As suggested, the IC₅₀ values of different treatments in Fig. 3f were listed in Supplementary Table 2.

Supplementary Table 2. Cytotoxicity (IC₅₀ values^{a)}) of MTHMS, MnCO+TSSI+L, THMS+L and MTHMS+L to 4T1 cells (CCK-8 assay).

Formulations	4T1 (µg mL ⁻¹)	
	TSSI	MnCO
MHMS	-	139.3
MnCO+TSSI+L	14.11	42.33
THMS+L	10.45	-
MTHMS+L	2.735	8.204

^{a)} Half maximal inhibitory concentration presented as equivalent concentrations of TSSI and MnCO.

Question 4: In Supplementary Fig. 8, the variation between sphere-like MTHMS group and virus-like MTHMS group in flow cytometric quantification of cellular uptake should be assessed through Student's t-test.

Response: We appreciate the reviewer's comments. As suggested, we assessed the variation between sphere-like MTHMS group and virus-like MTHMS group in flow cytometric quantification of cellular uptake through Student's t-test (Supplementary Fig. 9).

Supplementary Fig. 9. Flow cytometric quantification of 4T1 cells incubated with sphere-like MTHMS and virus-like MTHMS for 1 h and 4 h. Data represent the mean ± s.d. Statistical significance was calculated through two-tailed student's t-test.

Question 5: In Supplementary Fig. 14, the legends should be noted in figure to point out the specific treatment of groups.

Response: We agree with the reviewer's comments. We added the legends in Supplementary Fig.

23 to point out the specific treatment of groups.

Supplementary Fig. 23. Hepatorenal function parameters of 4T1 tumor-bearing mice after intravenous injection of different formulations. (a) ALT (U L⁻¹): alanine aminotransferase; AST (U L⁻¹): aspartate aminotransferase; (b) BUN (mmol L⁻¹): blood urea nitrogen; CREA (μmol L⁻¹): creatinine.

Question 6: There are many methods to detect CO, authors used hemoglobin here. Authors should provide some references to support their choice in this aspect.

Response: We appreciate the reviewer's comments. According to the reviewer's suggestion, the related references of hemoglobin assay for CO detection were added to support the choice in this aspect: *Adv Funct Mater.* 2019, 29, 1900095; *Adv Sci.* 2021, 8, 2004391; *Adv Mater.* 2015, 27, 6741-6746; *Biomaterials.* 2021, 274, 120894.

Question 7: In the experimental section of in vitro ROS generation assessment, authors indicated that "the fluorescence was detected through PL instrument at 525 nm". Authors should add the excitation wavelength for the measurement.

Response: We appreciate the reviewer's comments. As reviewer suggested, we added the excitation wavelength for the measurement in the revised manuscript: "The fluorescence at 525 nm was detected through PL instrument with excitation at 488 nm."

Question 8: The detailed statistical analysis methods should be individually indicated in every figure captions.

Response: We agree with the reviewer's comments. As reviewer suggested, the detailed statistical analysis methods were individually indicated in every figure captions.

Question 9: Authors should add the introduction and discussion about the current strategies of

cGAS-STING pathway activation in the field of nanomedicine.

Response: We appreciate the reviewer's comments. In the revised manuscript, we added the introduction and discussion about the current strategies of cGAS-STING pathway activation in the field of nanomedicine: "The development of nanomedicine-based strategies for cGAS-STING pathway activation profoundly revolutionized cancer immunotherapy (*Trends Pharmacol Sci.* 2022, 43, 957-972). Recently, some DNA-damaging drugs, such as teniposide, cisplatin, and olaparib (PARP inhibitor), have manifested the capability to activate cGAS-STING signaling in cancer cell, inducing robust antitumor immune responses. For example, Hou and coworkers designed a nanoactivator that could lead to the presence of DNA in cytosol and improve Mn^{2+} accumulation in cancer cells (*ACS Nano.* 2020, 14, 3927-3940). The nanoactivator was stable in the systemic circulation and activated to release doxorubicin (Dox) and Mn^{2+} to damage DNA and enhance cGAS-STING activity, which increased DC maturation and boosted intratumoral infiltration of cytotoxic T lymphocytes. Herein, we revealed the promising potential of tumor microenvironment-driven gas therapy in cGAS-STING pathway activation."

REVIEWERS' COMMENTS

Reviewer #1 (Remarks to the Author):

The corrections made are sufficient and good. The paper is now recommended for publication in Nat. Commun.

Reviewer #2 (Remarks to the Author):

The revised manuscript have addressed almost all the comments from the reviewers. So it can be accepted after minor revision.

1. In ref. 24, 29, 31, and 44, the name of the magazine should be abbreviate的, please corrected them.

Reviewer #3 (Remarks to the Author):

The authors have satisfactorily answered my questions.

One minor remark: legend to Fig. 25c: the tumor pictures are photographs, not graphs. This may be corrected.

Reviewer #4 (Remarks to the Author):

The author 's reply eliminated all my doubts, the quality of the manuscript was significantly improved, and the manuscript was recommended for publication.

Response to Reviewers' comments:

Reviewer #1:

The corrections made are sufficient and good. The paper is now recommended for publication in Nat. Commun.

Response: We feel great thanks for your positive comments and valuable suggestions to improve the quality of our manuscript.

Reviewer #2:

The revised manuscript have addressed almost all the comments from the reviewers. So it can be accepted after minor revision.

Comment 1. In ref. 24, 29, 31, and 44, the name of the magazine should be abbreviated, please correct them.

Response: We appreciate the reviewer's comments. As reviewer suggested, the names of the journal titles in ref. 24, 29, 31, and 44 were abbreviated.

Reviewer #3:

The authors have satisfactorily answered my questions.

One minor remark: legend to Fig. 25c: the tumor pictures are photographs, not graphs. This may be corrected.

Response: We appreciate the reviewer's comments. As reviewer suggested, we revised "graphs" to "photographs" in the legend of Fig. 25c.

Reviewer #4:

The author's reply eliminated all my doubts, the quality of the manuscript was significantly improved, and the manuscript was recommended for publication.

Response: We sincerely appreciate your positive comments and professional review work on our article.